# Dietary Tea Polyphenols Improve Growth Performance and Intestinal Microbiota Under Chronic Crowding Stress in Hybrid Crucian Carp

**DOI:** 10.3390/ani15131983

**Published:** 2025-07-05

**Authors:** Zhe Yang, Gege Sun, Jinsheng Tao, Weirong Tang, Wenpei Li, Zehong Wei, Qifang Yu

**Affiliations:** 1State Key Laboratory of Developmental Biology of Freshwater Fish, Hunan Normal University, Changsha 410081, China; 202303012@hunnu.edu.cn (Z.Y.); 17873680766@163.com (G.S.); taojinsheng01@163.com (J.T.); jxsdsmxytwr@163.com (W.T.); 13973571030@163.com (W.L.); zehongw@hunnu.edu.cn (Z.W.); 2Yuelushan Laboratory, Changsha 410128, China

**Keywords:** tea polyphenols, crucian carp, crowding stress, stocking density, lipid metabolism, intestinal microbiota

## Abstract

In high-density aquaculture, fish are under immense pressure due to crowding stress, resulting in slow growth and weakened immunity. To solve this problem, we tested whether adding natural green tea extracts (tea polyphenols) to fish feed could help hybrid crucian carp cope with overcrowding. We fed groups of fish different tea polyphenols doses (0, 100, 200, or 400 mg/kg of feed) in high-density cages holding five times more fish than low-density cages. After 60 days, fish receiving 200 mg/kg of tea polyphenols showed remarkable improvements; they grew faster than stressed fish without additives, developed stronger internal defenses against stress damage, processed fats more efficiently, and maintained healthier gut microbiota. By naturally boosting fish health in crowded conditions, tea polyphenols offer fish farmers a safe, eco-friendly alternative to antibiotics and chemicals. This approach could increase sustainable fish production to meet global food demands while ensuring better animal welfare.

## 1. Introduction

The global demand for protein has been steadily increasing in parallel with the growth of the world population. As a critical component of the food production system, aquaculture has made remarkable progress in recent decades [1]. Hybrid fish represent an essential segment of fish species and exhibit advantages such as rapid growth, strong immunity, high ecological adaptability, and enhanced tolerance during transportation [2]. In recent years, we successfully developed a new type of high-quality hybrid crucian carp [(*Carassius auratus cuvieri* ♀ × *Carassius auratus* red var ♂) ♀ × *Carassius auratus cuvieri* ♂], namely “Hefang crucian carp II” (HCC2), by integrating biotechnology with population breeding and sexual hybridization. Compared to its predecessor, Hefang crucian carp I (*Carassius auratus cuvieri* ♀ × *Carassius auratus* red var ♂, HCC1), HCC2 demonstrates a growth rate that is approximately 60% higher than HCC1, as evidenced by the final body weight (556 g vs. 380 g) after the first year of observation [3]. However, high stocking densities can exert various adverse effects on fish, affecting their health, behavior, and overall well-being. The optimal stocking density for HCC2 remains unclear, which poses challenges to both fish welfare and aquaculture productivity [4].

Intensification involves increasing the density of individuals, which necessitates greater management of inputs, increased waste production, and heightened risk of pathogen transmission [5]. Extensive research demonstrates that crowding stress markedly impairs the immune system of fish, thereby increasing their vulnerability to various diseases [6]. Chronic stress in fish is associated with the downregulation of key components within the hypothalamic–pituitary–inter-renal (HPI) axis, such as adrenocorticotropic hormone (ACTH) or corticosteroid receptors, resulting in a compromised cortisol response [7]. Furthermore, crowding stress induces significant alterations in gut microbiota composition and metabolic pathways in fish [8]. Therefore, when confronted with the dilemma between animal welfare and economic benefits, the adoption of plant-based feed additives may be a feasible nutritional strategy [9].

Tea polyphenols (TPs), also known as catechins, are the most biologically active components of tea, accounting for 30–42% of the dry weight of solids in brewed green tea [10]. TPs exhibit a wide range of pharmacological and biological activities, including antioxidant [11], anti-inflammatory [12], antimicrobial [13], and intestinal microbes regulation [14]. The sustainability and cost-effectiveness of TPs make them advantageous as feed additives for farm animals. TPs can partially replace synthetic antioxidants (e.g., butylated hydroxytoluene and butylated hydroxyanisole) and antibiotics, thereby significantly reducing environmental pollution caused by chemical residues in feed to soil and water bodies [15]. In addition, with the advancement of industrial extraction methods such as membrane separation and supercritical CO_2_ extraction, the production cost of tea polyphenols has decreased significantly and is now approximately twice that of the synthetic antioxidant butylated hydroxytoluene [16].

Recently, TPs have garnered considerable attention owing to their anti-stress properties [17]. TPs effectively protected grass carp against Flavobacterium columnare-induced gill injury by suppressing oxidative stress, apoptosis, and inflammation, while enhancing antioxidant enzyme activity and immune responses [18]. Similarly, dietary TPs in the diet effectively mitigated oxidative damage induced by ammonia exposure in juvenile Wuchang bream (*Megalobrama amblycephala*), primarily through the enhancement of their antioxidant defense mechanisms and immune functions [19]. Further, the supplementation TPs in oxidized fish-oil-based feed improves intestinal morphology, promotes the enrichment of beneficial gut microbiota, and optimizes hepatic nutrient metabolism in spotted sea bass (*Lateolabrax maculatus*) [20]. Moreover, TPs restructured the intestinal microbiota composition in loaches (*Paramisgurnus dabryanus*) under chronic ammonia nitrogen stress, specifically enriching metabolic pathways linked to amino acid and nucleotide biosynthesis, thereby providing a potential mechanism for microbial-mediated adaptation to environmental stressors [21]. Owing to their broad spectrum of biological activities, TPs have been utilized as a promising feed additive in aquaculture species. However, the effects of TPs on mitigating chronic crowding stress in the valuable but understudied HCC2, employing a comprehensive approach that links physiological responses (growth, metabolism, and antioxidant defense) with intestinal microbiota remodeling, remain unexplored.

Therefore, this study was designed to systematically investigate whether dietary TPs supplementation can effectively alleviate chronic crowding stress in HCC2 and elucidate the underlying mechanisms. We hypothesize that TPs exert protective effects through enhancing antioxidant capacity, regulating lipid metabolism, and modulating the gut microbiome. By integrating microbiome analysis, we specifically aimed to evaluate how crowding stress and TPs intervention reshape intestinal microbial diversity and composition, assessing its potential influence on host health and stress resilience. This research provides insights into TPs’ action against crowding stress and offers robust scientific support for the application of this natural, plant-based additive as a sustainable strategy to improve fish welfare and productivity in high-density aquaculture systems.

## 2. Materials and Methods

### 2.1. TPs and Experimental Diets

To meet the nutritional requirements for crucian carp HCC2, four formulated diets were systematically designed (Table 1). The CON and CS groups were fed basal diets without TPs, whereas the three treatment groups (CSLTP, CSMTP, and CSHTP) received dietary supplementation with 100, 200, and 400 mg/kg TPs, respectively. The TPs material (purity > 98%), sourced from the National Engineering Research Center for Botanical Functional Ingredients Utilization at Hunan Agricultural University (Changsha, China), was composed primarily of catechins, with epigallocatechin gallate (EGCG) constituting >50% of the total polyphenols. Other major components included epicatechin, epicatechin gallate, epigallocatechin, gallocatechin gallate, catechin, gallocatechin, and gallic acid, as detailed in Appendix A.

During feed processing, raw ingredients underwent ultrafine pulverization through an 80-mesh sieve to ensure uniformity. The homogenized mixture was then conditioned with water and extruded into buoyant pellets (1 mm diameter) using single-screw extrusion equipment (Model EF-003, Huolibao Feed Co., Zhuhai, China). This method optimized pellet integrity and water stability, critical for crucian carp feeding behavior. Post-extrusion, all diets were dehydrated at 65 °C in a forced-air drying chamber and stored at −20 °C to preserve nutrient stability until feeding trials.

### 2.2. Fish and Experimental Conditions

The fish involved in this study were used in strict accordance with the recommendations in the Guidelines for the Care and Use of Laboratory Animals of the National Advisory Committee for Laboratory Animal Research in China and approved by the Animal Care Committee of Hunan Normal University (Permit Number: 4637). The experimental fish HCC2 (mean initial body weight: 2.01 ± 0.01 g) were sourced from the State Key Laboratory of Developmental Biology of Freshwater Fish, Hunan Normal University (Changsha, China). Prior to experimentation, fish underwent a 7-day acclimation phase under natural conditions.

After acclimation, a total of 6300 fish were randomly distributed into five experimental groups, each with three replicates. Each replicate was housed in a cage (1.5 × 1.5 × 1.0 m^3^) for a 60-day trial period. The control (CON) group had a low stocking density of 44.4 fish/m^3^, while the high stocking density in the crowding stress groups (CS, CSLTP, CSMTP, and CSHTP) was set at 222.2 fish/m^3^. Throughout the trial, the ponds were regularly disinfected and water quality conditioned, and the physicochemical factors of the pond water environment, dissolved oxygen (>5.00 mg/L), water temperature (30.0 ± 2 °C) and pH (7.16–7.38), and unionized ammonia nitrogen (<0.001 mg N/L), were monitored using a Hach HQ40d multiparameter meter (Loveland, CO, USA).

### 2.3. Harvesting and Sample Collection

Sample collection procedures were performed as previously conducted by our laboratory [22]. Upon completion of the trial, fish were removed from each group. The fish were anesthetized using MS-222 (Merck, Darmstadt, Germany). Blood was collected from the caudal vein and immediately centrifuged at 3000× *g* for 10 min at 4 °C. In each replicate cage, the serum from three fish was pooled and mixed equally (resulting in a total of six composite serum samples per group). The fish after blood sampling were immediately placed on ice. Their livers and intestines were subsequently dissected and separately stored in 1.5 mL enzyme-free tubes for the assessment of antioxidant-related indicators and qPCR analysis. All collected samples were rapidly frozen in liquid nitrogen and subsequently stored at −80 °C for long-term preservation. For histological examination, the liver tips of several additional fish were carefully excised and fixed in 4% paraformaldehyde solution.

### 2.4. Growth Performance

At the beginning and end of the experiment, the average initial body weight (IBW) and average body final weight (FBW) of each group were recorded. The weight gain rate (WGR), specific growth rate (SGR), feed conversion ratio (FCR), survival rate (SR), and condition factor (CF) were determined using the following formulas:Weight gain rate (WGR), % = 100 × (FBW − IBW)/IBWSpecific growth rate (SGR), %/d = 100 × (Ln FBW − Ln IBW)/daysFeed conversion ratio (FCR), % = (FBW − IBW)/average total feed intakeSurvival rate (SR), % = 100 × (final number/initial fish number)Condition factor (CF), g/cm^3^ = 100 × [FBW/(body length)^3^]

### 2.5. Serum Biochemical Indices

Serum levels of energy metabolism indicators [lactate dehydrogenase (LDH), glucose (GLU), and lactate (LACT)], lipid metabolism indicators [triglycerides (TG), total cholesterol (TC), non-esterified fatty acids (NEFA), low-density lipoprotein cholesterol (LDL-C), and high-density lipoprotein cholesterol (HDL-C)], and immune-related indicators [albumin (ALB), alanine aminotransferase (ALT), aspartate aminotransferase (AST), and alkaline phosphatase (ALP)] were measured using a Roche cobas-c-311 fully automatic biochemical analyzer (Sandhofer Strasse, Mannheim, Germany).

### 2.6. Antioxidant Capacity

The frozen liver samples were thawed at 4 °C. Subsequently, 100 mg of liver or intestines was transferred to 1.5 mL microcentrifuge tubes. Each tube was then supplemented with 1 mL of extraction solution. The antioxidant-related parameters [total antioxidant capacity (T-AOC, ABTS reduction method), superoxide dismutase (SOD, WST-8 inhibition method), catalase (CAT, Molybdate assay method), and glutathione (GSH, DTNB reaction method)] in serum and liver tissue were measured according to the protocols provided by Beijing Boxbio Science & Technology Co., Ltd. (Beijing, China).

### 2.7. mRNA Expression

Total RNA was extracted from the liver and intestine tissue of HCC2 using RNA isolator Total RNA Extraction Reagent (Vazyme, Nanjing, China). The quality and purity of the total RNA were assessed by spectrophotometry, with an acceptable OD 260/280 ratio ranging from 1.8 to 2.2, indicating suitability for real-time quantitative PCR (RT-qPCR). Subsequently, the extracted RNA was reverse transcribed into cDNA using the HiScript II Q RT SuperMix for qPCR (Vazyme) according to the manufacturer’s protocol. Specific primer sequences are listed in Table 2, with *β-actin* serving as the internal reference gene. RT-qPCR was conducted on a QuantStudioTM 5 Real-Time PCR System (Thermo Fisher Scientific, Waltham, MA, USA) with a total reaction volume of 20 μL. The thermal cycling conditions were as follows: initial denaturation at 95 °C for 30 s, followed by 40 cycles of denaturation at 95 °C for 15 s and annealing/extension at 60 °C for 1 min. All primers for RT-qPCR were designed using Premier 5.0 software and synthesized by Beijing Tsingke Biotech Co., Ltd., Beijing, China. The relative expression levels of target genes were analyzed using the 2^−ΔΔCt^ method [23].

### 2.8. Intestinal Microbiota Analysis

The microbiomics approach is described in detail in the Appendix A. Briefly, DNA was extracted from the intestinal contents of hybrid crucian carp HCC2 using the E.Z.N.A.^®^ Soil DNA Kit (Omega Bio-tek, Norcross, GA, USA). The V3-V4 hypervariable region of bacterial 16S rRNA genes was amplified with universal primers (338F/806R) and sequenced on the Illumina MiSeq platform (Illumina, San Diego, CA, USA). Raw sequencing reads were quality-filtered, merged, and processed using fastp and FLASH, followed by clustering into operational taxonomic units (OTUs) at a 97% similarity threshold via UPARSE. Taxonomic annotations were performed against the Silva 16S rRNA gene database (v138). Chloroplast and mitochondrial sequences were removed to refine the dataset. Subsequently, α-diversity indices were calculated to evaluate microbial richness and evenness, while principal component analysis (PCA), principal co-ordinates analysis (PCoA), and partial least squares discriminant analysis (PLS-DA) were conducted to assess the similarity of microbial community structures among samples. Additionally, LEfSe analysis was employed to identify bacterial taxa that significantly differed in abundance between groups at various taxonomic levels (LDA score > 3.0).

### 2.9. Statistical Analysis

Biochemical data (means ± SEM) were subjected to one-way ANOVA analysis. Outliers were identified using Grubbs’test (α = 0.05) and excluded. Data were log-transformed where necessary to meet normality assumptions for ANOVA. Following the verification of the homogeneity of variances, the Duncan’s multiple-range test was conducted for pairwise comparisons. Differences with *p*-Values less than 0.05 were deemed statistically significant. All graphical representations were generated using GraphPad Prism software version 10.4 (San Diego, CA, USA). Statistical analyses were performed using SPSS version 24.0 (IBM Corporation, Armonk, NY, USA).

Bioinformatic analysis of the gut microbiota was carried out using the Majorbio Cloud platform (https://cloud.majorbio.com, accessed on 5 December 2024). Based on the OTUs information, rarefaction curves and alpha diversity indices, including observed OTUs, Chao1 richness, Shannon index, observed richness, and Good’s coverage, were calculated with Mothur v1.30.1. The similarity among the microbial communities in different samples was determined by principal coordinate analysis (PCoA) based on Bray–Curtis dissimilarity using Vegan v2.5-3 package. The PERMANOVA test was used to assess the percentage of variation explained by the treatment along with its statistical significance using Vegan v2.5-3 package. The linear discriminant analysis (LDA) effect size (LEfSe) (http://huttenhower.sph.harvard.edu/LEfSe, accessed on 22 December 2024) was performed to identify the significantly abundant taxa (phylum to genera) of bacteria among the different groups (LDA score > 3.0).

## 3. Results

### 3.1. Effects of TPs Supplementation on Growth Performance

After 60 days of cultivation, the effects of TPs on the growth performance of hybrid crucian carp HCC2 are summarized in Table 3. Compared with the CON group, the crowding stress conditions (CS group) significantly decreased FBW, WGR, SGR, and FE (*p* < 0.05), while significantly increasing FCR (*p* < 0.05). No significant differences were observed in SR or CF (*p* > 0.05). Compared with the CS group, the CSMTP group exhibited a significantly lower CF (*p* < 0.05). Although TPs supplementation tended to increase WGR, SGR, FE, and FBW and decrease FCR and feeding rate (FR), these differences were not statistically significant (*p* > 0.05).

### 3.2. Effects of TPs Supplementation on Serum Biochemical Indices

The serum biochemical indices of hybrid crucian carp HCC2 are presented in Table 4. Compared with the CON group, the CS group significantly increased serum LDH and TG (*p* < 0.01), as well as ALB (*p* < 0.05), while significantly decreasing serum GLU, LDL-C, ALT, AST, and ALP (*p* < 0.05). However, crowding stress conditions had no significant effects on LACT, TC, NEFA, or HDL-C.

Dietary TP groups significantly decreased LDH and GLU (*p* < 0.01), significantly reduced LACT (*p* < 0.05), and significantly increased ALP (*p* < 0.05). The CSLTP group significantly decreased TG and ALT (*p* < 0.01) and significantly increased AST (*p* < 0.05). Both the CSLTP and CSMTP groups significantly increased NEFA (*p* < 0.05), whereas the CSMTP and CSHTP groups significantly decreased HDL-C (*p* < 0.05) and markedly increased ALT (*p* < 0.01). All TPs-supplemented groups had no significant effects on TC, LDL-C, TP, or ALB (*p* > 0.05).

### 3.3. Effects of TPs Supplementation on Antioxidant Capacity

The results of the determination of antioxidant capacity and intestinal digestive enzyme in hybrid crucian carp HCC2 are listed in Table 5. Compared with the control group (CON) under low-density rearing conditions, the oxidative stress level in the crowded stress group (CS) under high-density rearing conditions significantly increased, as evidenced by elevated serum T-AOC (*p* < 0.05) and hepatic SOD levels (*p* < 0.05), along with reduced GSH content (*p* < 0.05). After dietary supplementation with TPs, the degree of oxidative stress was alleviated and related antioxidant indicators returned to normal levels. However, compared with CS groups, serum SOD, CAT, and GSH, as well as hepatic CAT, showed significant increases (*p* < 0.05), indicating that TPs effectively improved the overall antioxidant capacity of HCC2.

### 3.4. Effects of TPs Supplementation on mRNA Expression

As shown in Figure 1, compared with the control group (CON), the crowding stress group (CS) exhibited significantly increased expression of *SOD*, *CPT1*, and *dgat2* in the liver (*p* < 0.05), while GPX expression was significantly reduced (*p* < 0.05). Additionally, crowding stress did not significantly affect the expression of *CAT*, *nrf2*, *acox1*, or *LPL1* in the liver (*p* > 0.05). The TPs-supplemented groups significantly enhanced the expression of *CAT* and *acox1* in the liver (*p* < 0.05). Moreover, both the CSLTP and CSMTP groups markedly increased the expression of *SOD*, *nrf2*, and *gdat2* in the liver (*p* < 0.05).

### 3.5. Microbial Sample Information and Composition Analysis

At the designated 97% similarity threshold, 1,788,358 high-quality sequences were clustered into 5155 operational taxonomic units (OTUs) from 25 samples (*n* = 5). Pan/Core species analysis is designed to characterize the trends in the total species count and core species abundance as sample size expands. As shown in Appendix A, the Pan/Core curves indicated a sufficient number of OTUs across all groups for subsequent analyses.

Figure 2 presents the results of the α-diversity index analyzed using the Kruskal–Wallis H test. As shown in Figure 2A, the Sobs index demonstrates significant differences among various treatment groups (*p* < 0.05), with the CSHTP group exhibiting a marked reduction in microbial richness (*p* < 0.05). Additionally, Supplementary Figure 2 illustrates the rarefaction curve based on the Sobs index, which tends to plateau, indicating that the sequencing depth is sufficient to capture the microbial community composition within the samples. Furthermore, Figure 2B displays the results of the Shannon index analysis. While a statistically significant difference was observed between the CON and CS groups (*p* < 0.05), no significant differences were detected among the treatment groups (*p* > 0.05). Similarly, the Chao index in Figure 2C aligns closely with the findings of the Sobs index, revealing a statistically significant difference (*p* < 0.05) between the CSHTP and CS groups. In contrast, the Coverage index in Figure 2D shows no significant differences among the groups (*p* > 0.05).

Venn diagrams (Figure 3) were used to visualize the shared and unique species across multiple groups. Additionally, Appendix A presents a detailed breakdown of the shared and unique species at the OTU level. The total number of OTUs increased from 2182 in the CON group to 2576 in the CS group, while the number of unique OTUs decreased significantly from 415 to 349. Among all groups, the CSMTP group had the highest total OTU count (2653), whereas the CSHTP group exhibited the lowest (2052). Only 1061 core OTUs were shared among all five groups. Notably, the number of unique OTUs in the treatment groups increased (CSLTP: 137; CSMTP: 349; and CSHTP: 359), particularly in the medium- and high-dose groups.

### 3.6. Comparative Analysis of Microbiota

As shown in Figure 4A, PCA educes dimensions to display the overall differences among samples. Although the first principal component (PC1) and PC2 explain 19.24% and 15.35% of the variance, respectively, the statistical test for the differences between groups (*p* = 0.105) did not reach significance (*p* > 0.05). Meanwhile, Figure 4B displays the results of the PCoA, which performed to visualize and quantify differences among samples (based on the Bray–Curtis distance). Despite the relatively low proportion of variance explained by inter-group differences (R^2^ = 7.02%), there was a significant separation among the groups (*p* < 0.05). However, PLS-DA is a supervised analysis method that requires grouping the detection samples by category, thereby emphasizing group differences. In Figure 4C, each group discriminated from the other groups, suggesting distinct microbial community characteristics in the control group compared to the treatment groups. Finally, we conducted an inter-group difference analysis of β-diversity based on the distance index between samples within the group. The results (Figure 4D) showed that there were extremely significant differences in the distribution of distance values among the groups (*p* < 0.001).

The community bar plot analysis serves as the primary method for elucidating the microbial community structure within the samples. The bar chart visually represents the microbial communities across all groups, facilitating a comparison of the changing trends in phylum and genus abundance. At the phylum level (Figure 5A), *Proteobacteria* and *Firmicutes* were the dominant taxa across all groups, collectively representing over 50% of the microbiota. In the CON group, *Firmicutes* accounted for approximately 36.6% of the community, while *Proteobacteria* constituted 29.0%. However, under crowding stress (CS group), the relative abundance of *Proteobacteria* increased to 37.1%, whereas *Firmicutes* decreased to 12.9%. In the treatment groups (CSLTP, CSMTP, and CSHTP), the proportion of *Firmicutes* partially recovered to 14.0–21.6%, while the abundance of *Proteobacteria* remained largely unchanged compared to the CS group. With regard to minor phyla, including *Actinobacteriota*, *Bacteroidota*, and *Cyanobacteria*, these remained at low abundances (<10% each) with minimal variation among groups. The genus-level analysis (Figure 5B) revealed distinct shifts in microbial composition. The unclassified genus *norank_f_Rhizobiales_Incertae_Sedis* was the most abundant taxon in all groups, representing 25% in the CON group, increasing to 35% under crowding stress (CS group), and declining to 20% in the CSHTP group. Interestingly, the genus *Lactococcus* dominated in the CON group, but the stress-sensitive genera like *unclassified_c_Bacilli* and *Pantoea* nearly disappeared in the CS group but reappeared at 2–5% in treatment groups. Additionally, the Kruskal–Wallis H test for significance of differences between groups is listed in Appendix A.

## 4. Discussion

In the process of intensive aquaculture, increasing stocking density to achieve high yield and economic benefit is often inevitable; however, this approach may not always be reasonable or sustainable. Stressors typically challenge various physiological functions as fish strive to restore their homeostatic status. Moreover, enhancing the host’s innate immune mechanisms, health condition, and growth rate represents an ideal approach for disease and stress management in modern aquaculture [24]. Therefore, it is essential to find appropriate nutritional strategies to promote fish growth and improve their welfare [25]. Our study systematically revealed for the first time that TPs alleviate chronic crowding stress in hybrid crucian carp HCC2 via multifaceted mechanisms, including antioxidant enhancement, lipid metabolism regulation, and intestinal microbiota modulation.

### 4.1. Effects on Growth Performance

Under crowding stress, the growth and development of individuals experience a delay, concomitantly with a diminution in immune function. Additionally, such organisms are compelled to allocate increased energy to satisfy the escalated metabolic demands, consequently leading to an inhibition of somatic growth [26]. In terms of growth performance, high-density rearing significantly suppressed the final body weight (FBW decreased by 37.4%), weight gain rate (WGR decreased by 42.4%), and specific growth rate (SGR decreased by 43.4%) of HCC2 (Table 3), which is consistent with the density-dependent growth inhibition observed by Chen et al. in Gibel carp (*Carassius auratus gibelio*) [27]. Notably, CSLTP group increased the specific growth rate by 50.6% (from 12.5% to 18.82%/d), and this improvement was significantly better than the growth-promoting effect of phenolic compound (quercetin) in common carp reported by Ghafarifarsani et al. [9]. Indeed, the growth-promoting potential of TPs has been consistently underestimated. Extensive research has demonstrated that TPs could significantly improve the growth performance of juvenile fish, including hybrid grouper [28], hybrid sturgeon [29], and largemouth bass [30]. Moreover, it is noteworthy that neither the stress conditions nor the TPs treatment had a significant impact on the survival rate. This finding implies that the existing stress levels did not result in a marked increase in fish mortality, and the administered dose of TPs did not adversely affect fish survival.

### 4.2. Effects on Serum Biochemical Indices

When crowding stress disrupts the homeostasis of fish, compensation or adaptive mechanisms are activated to counteract harmful environmental factors. However, prolonged exposure to stress may cause the stress response to lose its adaptive value and immune protection function, thereby disrupting normal physiological conditions and negatively impacting the growth, development, and reproduction of fish [31]. Crowding stress significantly shifts energy metabolism toward anaerobic pathways, as indicated by increased serum LDH and TG levels in high stocking density groups, reflecting enhanced glycolytic activity and abnormal lipid accumulation [32]. Conversely, TPs intervention demonstrates dose-dependent improvements, reduced LDH levels alongside upregulated hepatic CPT1 expression activate the PPARα signaling pathway, enhancing fatty acid β-oxidation. This aligns with findings from studies on juvenile tilapia (*Oreochromis niloticus*) fed a high-fat diet supplemented with TPs [33]. Additionally, TPs increase ALP activity, potentially improving enterohepatic lipoprotein assembly and liver function, while stabilizing stress-induced elevations in AST and ALT levels, thus reducing hepatocyte membrane permeability and cellular damage. In summary, lipid metabolism rebalancing is achieved through increased HDL-C and NEFA levels, along with decreased LDL-C, optimizing insulin sensitivity and reverse cholesterol transport. The synergistic effects of TPs—reducing anaerobic burden via LDH/GLU modulation, promoting lipid oxidation via CPT1-PPARα activation, and enhancing hepatic detoxification via ALP-mediated pathways—collectively restore metabolic homeostasis. These findings are consistent with our prior study on TPs alleviating acute heat-stress-induced death of HCC2 through modified lipid metabolisms in the liver [34].

### 4.3. Effects on Antioxidant Capacity

Under crowding stress, fish exhibit stress responses and rely on the antioxidant system to scavenge free radicals, thereby maintaining homeostasis [35]. However, when prolonged external stimuli exceed the capacity of the antioxidant system, oxidative stress ensues, resulting in tissue damage. Especially, T-AOC, SOD, CAT, and GSH serve as critical biomarkers for assessing oxidative stress in fish. Analysis of the antioxidant system revealed that, in the CS group, the SOD activity in the liver increased by 17.5%, while the GSH content decreased by 39.2% (Table 5). This compensatory antioxidant response is consistent with the oxidative stress characteristics found in the long-term crowding stress of grass carp (*Ctenopharyngodon idella*) [6]. After TPs supplementation, the serum CAT activity increased by 48.9%, and the expression of the nrf2 gene in the liver was upregulated by 2.3 times (Figure 1), confirming that it enhances the expression of phase II detoxification enzymes by activating the Nrf2/Keap1 pathway. Similarly, Mi et al. found that adding dietary (-)-Epicatechin (one kind of TPs) regulates antioxidant status by the same mechanism in Yellow River carp (*Cyprinus carpio*) [36]. In particular, the expression of the *acox1* gene of TPs-supplemented groups increased by at least 2.5 times, suggesting the synergistic activation of the peroxisomal fatty acid oxidation pathway in hybrid crucian carp HCC1 [37]. Therefore, future studies should quantify lipid peroxidation products (such as malondialdehyde) in TP-supplemented fish.

### 4.4. Effects on Gut Microbiota

#### 4.4.1. Microbial Richness and Diversity Changes

The gut microbiota plays a vital role in the physiological and metabolic processes of the host [38]. The Sobs index, which directly measures species richness by indicating the observed number of OTUs, demonstrates that crowding stress or TP treatments significantly reduce species count in intestinal microbiota (Figure 2A). While the Shannon index evaluates both species richness and evenness (Figure 2B), its analysis reveals that, although the Sobs index exhibited substantial fluctuations, microbial community evenness (i.e., the equilibrium of species relative abundances) remained largely unaffected, maintaining overall α-diversity stability. Complementary to these findings, the Chao index (Figure 2C) estimates total species richness—including undetected rare species—and aligns with Sobs index trends, further supporting the observed reduction in microbial diversity under specific treatments. In addition, the Coverage index (Figure 2D) reflects sequencing depth adequacy, with values approaching 1 (indicating comprehensive species detection) and consistent sequencing depth across treatment groups, thereby validating result reliability.

Furthermore, the Venn diagrams (Figure 3) indicate that crowding stress altered the distribution of OTUs in the intestinal microbiota of hybrid crucian carp, suggesting stress-induced microbial dysbiosis and reduced diversity of the original microbiota [39]. However, the persistence of 1061 core OTUs shared across all experimental groups implies that essential microbiota responsible for maintaining fundamental intestinal functions remained resilient under both stress and treatment conditions, consistent with observations in crucian carp (*Carassius auratus*) under environmental perturbations [40]. Notably, the increased unique OTUs observed in treatment groups—particularly the medium-dose (CSMTP: 349) and high-dose (CSHTP: 359) groups—demonstrate a potential dose-dependent response, which may reflect either successful colonization of new microbial species or partial restoration of the original microbial community through therapeutic intervention [41].

In summary, these findings suggested that crowding stress altered gut microbial richness and composition in hybrid crucian carp HCC2. A similar observation has been reported by Yang et al., demonstrating a significant increase in the abundance of gut microbiota in both the foregut and hindgut of largemouth bass (*Micropterus salmoides*) after exposure to acute crowding stress [42]. This finding highlights the potential impact of stress on gut microbial communities in aquatic organisms. However, researchers have conducted in-depth analyses of the interactions between gut microbes and polyphenols, a process that leads to the production of a variety of metabolites of physiological relevance [43]. An increasing number of evidence demonstrates that the primary mechanism by which TPs enhance bodily health is through maintaining the balance of gut microbiota [44]. A further study by Chai et al. demonstrated that TPs restructured the intestinal microbiota composition of loaches (*Paramisgurnus dabryanus*) under chronic ammonia nitrogen stress, with specific enrichment of metabolic pathways associated with amino acid and nucleotide biosynthesis, revealing a potential microbial-mediated mechanism for environmental stress adaptation [21]. The reduced α-diversity directly corresponded to β-diversity dispersion, indicating that TPs not only increased species richness but fundamentally restructured community assembly under stress.

#### 4.4.2. Community Structure and Inter-Group Variation

The β-diversity inter-group difference analysis refers to the differences between samples. Statistical tests are conducted on the distance differences between different groups to reflect the degree of dispersion of samples within groups [45]. Although the microbial communities of each group did not show significant separation in the PCA analysis, the PCoA analysis could better capture the nonlinear relationships, thereby supporting the existence of certain differences between the treatment groups and the control groups. However, the inter-group variation accounted for only 7% of the total variation, indicating that other factors (such as individual differences) might have a greater impact on the structure of the microbial communities. The PLS-DA results illustrate distinct microbial community characteristics in the control group compared to the treatment groups. The between-group separation of PLS-DA and the significance of the β-diversity test support each other. Based on the results above, we hypothesize that crowding stress may lead to alterations in the structure of the intestinal microbiota in hybrid crucian carp HCC2. A previous investigation showed that the gut microbiome of *Megalobrama amblycephala* plays a dominant role in the adaptation to crowding stress, and the abundance of microbes was significantly changed [46]. Therefore, the specific microbial alterations serve as a critical factor in the mechanism by which TPs mitigate crowding stress.

In Figure 5, *Proteobacteria* and *Firmicutes* dominated across all groups. Their prominence suggests a stress-adapted microbial community, as both phyla are commonly associated with metabolic flexibility and host–microbe interactions under environmental perturbations [47]. *Proteobacteria* is known to encompass pathogens, such as *Vibrio alginolyticus*, *Aeromonas hydrophila*, and *Pseudomonas*. The high abundance of *Proteobacteria* observed in the CS group may suggest potential gut dysbiosis or immune system activation in fish [48]. However, numerous studies have shown that TPs exhibit significant inhibitory effects on pathogenic bacteria of the *Proteobacteria* in aquatic animals. For instance, the supplementation of 120 mg/kg TPs could effectively alleviate the intestinal barrier function injury induced by *Aeromonas hydrophila* in grass carp (*Ctenopharyngodon idella*) [49]. Further, *Firmicutes* linked to energy harvest and short-chain fatty acids (SCFAs) production, and their dominance could reflect enhanced energy metabolism to cope with crowding stress [50]. Similarly, TPs promote the metabolic activity of *Firmicutes* (especially *Lactococcus*) and *Bacteroidetes*, increasing the production of SCFAs. These metabolic products may enhance intestinal barrier function and inhibit the release of inflammatory factors [51].

## 5. Conclusions

This study demonstrates that dietary supplementation with TPs effectively alleviates chronic crowding stress in hybrid crucian carp HCC2. Mechanistically, TPs enhance antioxidant defense by activating the Nrf2/Keap1 pathway, increasing the activities of antioxidant enzymes, elevating glutathione levels, and mitigating oxidative stress. Simultaneously, TPs regulate lipid metabolism through PPARα signaling activation, upregulating fatty acid β-oxidation genes, thereby improving energy homeostasis. Furthermore, TPs ameliorate stress-induced gut microbiota dysbiosis by reducing *Proteobacteria* proliferation while restoring *Firmicutes* abundance, consequently diminishing stress-associated pathogenic risks. These findings show dietary TPs supplementation as a scientifically validated strategy for enhancing stress resilience and ensuring sustainable intensive aquaculture of hybrid crucian carp. Based on both biological effectiveness and economic viability, 200 mg/kg was determined to be the optimal additive dosage.

## Figures and Tables

**Figure 1 animals-15-01983-f001:**
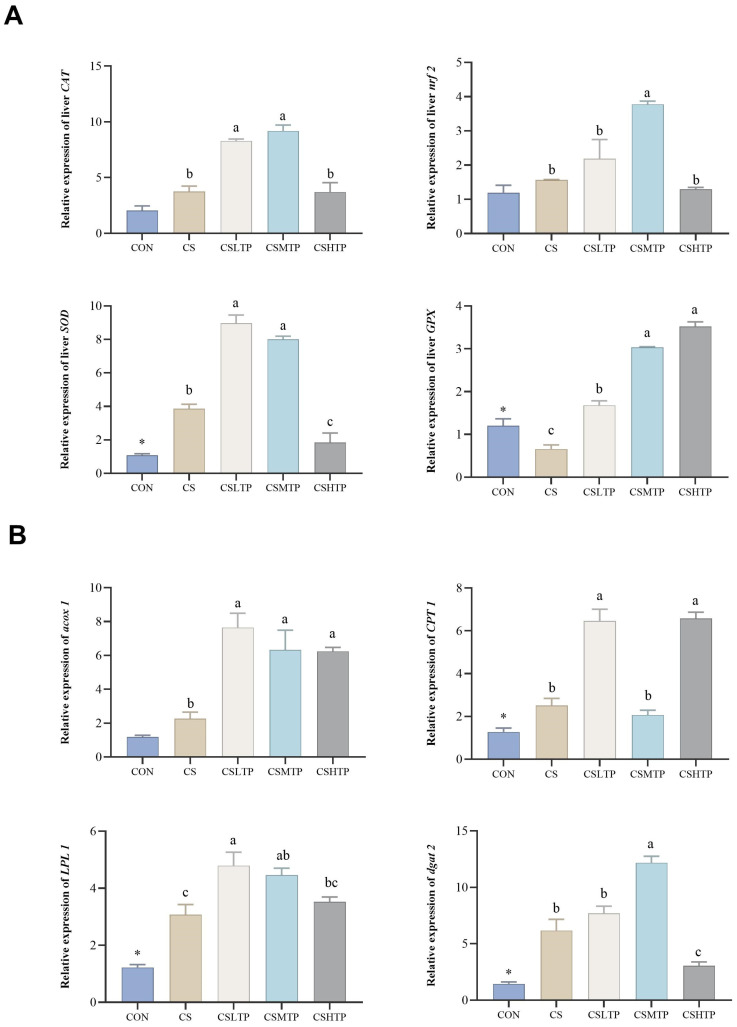
Effects of TPs under crowding stress conditions on the expression of liver genes involved in antioxidant defense (**A**) and lipid metabolism (**B**) in hybrid crucian carp HCC2. Data are shown as the AVG ± SEM (*n* = 3). The asterisk (*) as a superscript indicates a significant difference between CON and CS (*p* < 0.05). Different superscript letters signify a significant difference between CS and CSLTP, CSMTP, and CSHTP groups (*p* < 0.05). Abbreviation: *CAT*: Catalase, *nrf2*: Nuclear Factor Erythroid 2-Related Factor 2, *SOD*: Superoxide Dismutase, *GPX*: Glutathione Peroxidase, *dgat2*, Diacylglycerol O-Acyltransferase 2. *acox1*: Acyl-CoA Oxidase 1, *CPT1*: Carnitine Palmitoyl Transferase 1, *LPL1*: Lipoprotein Lipase 1.

**Figure 2 animals-15-01983-f002:**
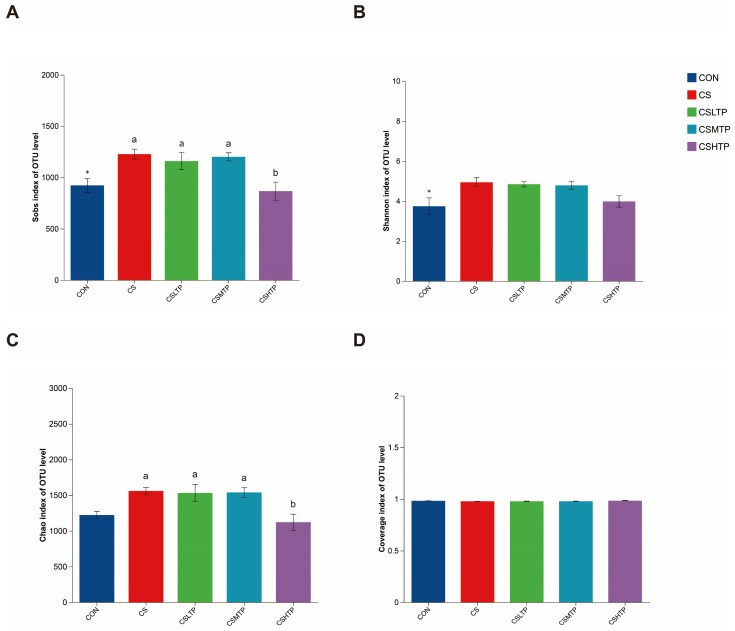
Kruskal–Wallis H test for the Sobs (**A**), Shannon (**B**), Chao (**C**), and Coverage (**D**) indices of the hybrid crucian carp intestinal microbiota in response to crowding stress at the OTU level. The horizontal axis denotes the group names, while the vertical axis indicates the average value of the index for each group. Data are shown as the AVG ± SEM (*n* = 6). The asterisk (*) as a superscript indicates a significant difference between CON and CS (*p* < 0.05). Different superscript letters signify a significant difference between CS and CSLTP, CSMTP, and CSHTP groups (*p* < 0.05).

**Figure 3 animals-15-01983-f003:**
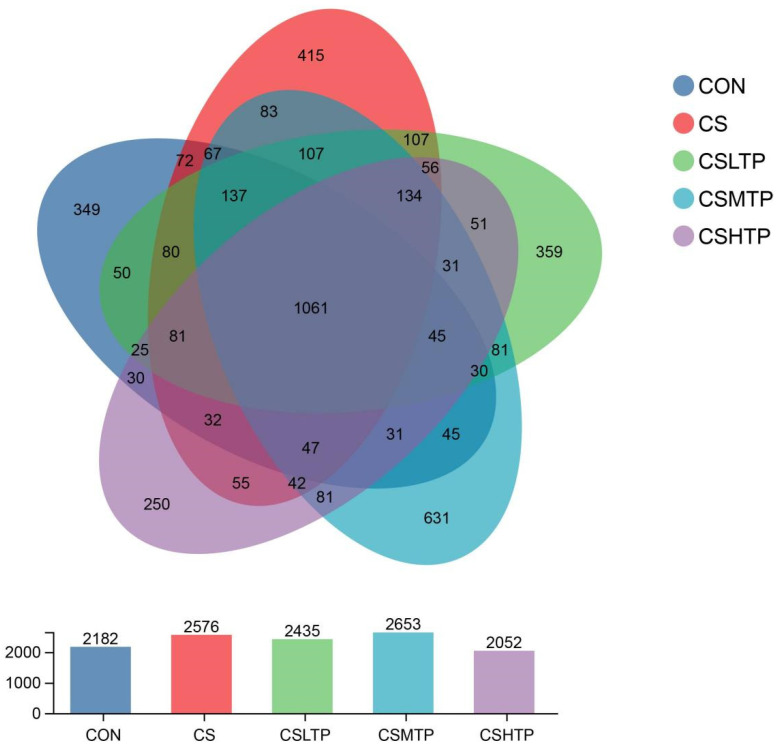
The Venn diagrams of the hybrid crucian carp intestinal microbiota in response to crowding stress at the OTU level. Different colors are used to distinguish different groups. The numbers in the overlapping areas represent the number of species shared by multiple groups, while the numbers in the non-overlapping areas indicate the number of species unique to each group.

**Figure 4 animals-15-01983-f004:**
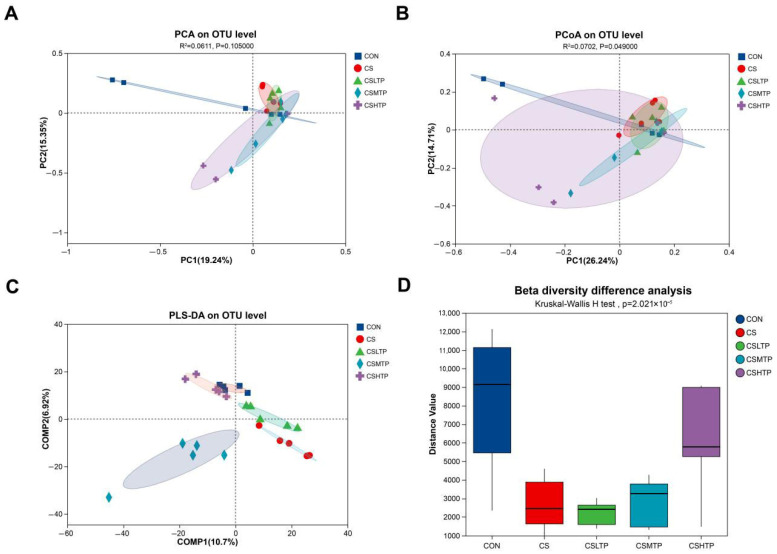
The PCA (**A**), PCoA (**B**), PLS-DA (**C**), and β-diversity inter-group difference analysis (**D**) in the hybrid crucian carp intestinal microbiota in response to crowding stress at the OTU level. (**B**): The X-axis and Y-axis correspond to the two selected principal component axes, with the percentages reflecting the proportion of variance explained by these components. Points distinguished by color or shape denote samples from different groups, where closer proximity between points signifies greater similarity in species composition. (**C**): Points with different colors or shapes represent sample groups under distinct environmental or experimental conditions. Comp1 and Comp2 indicate the potential influencing factors contributing to the deviation in microbial composition among samples. (**D**): The Kruskal–Wallis H test was used to analyze the differences in β-diversity among different groups. Each box plot in the figure shows the distribution of Bray–Curtis distance values between each pair of samples within the group.

**Figure 5 animals-15-01983-f005:**
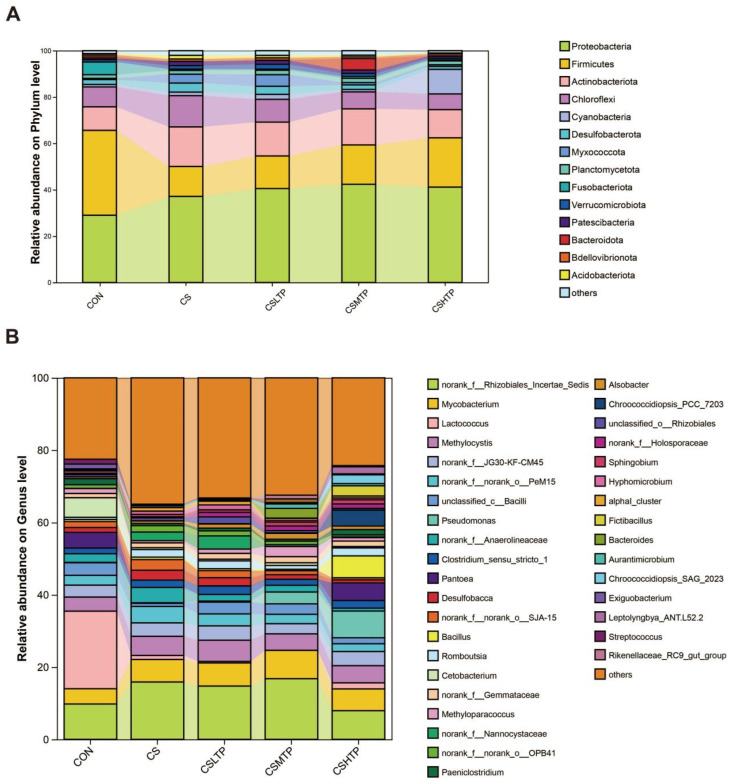
The community bar plot analysis in the hybrid crucian carp intestinal microbiota in response to crowding stress at the phylum (**A**) and genus (**B**) level. The X-axis represents the group names, and the Y-axis shows the proportion of each species within the group. Different colors of the bars represent different species, and the length of the bars indicates the proportion of the species.

**Table 1 animals-15-01983-t001:** Formulation and nutrient composition of the experimental diets ^a^ (%, dry matter).

Ingredients	Experimental Groups Diets				
	CON	CS	CSLTP	CSMTP	CSHTP
Wheat Flour	29.7	29.7	29.69	29.68	29.66
Soybean Meal	27	27	27	27	27
Rapeseed Meal	15	15	15	15	15
Peanut Hulls	12	12	12	12	12
Fish Meal	10	10	10	10	10
Fish Oil	2	2	2	2	2
Ca(H_2_PO_4_)_2_	1.5	1.5	1.5	1.5	1.5
Soybean Oil	1	1	1	1	1
Additive Premixes ^b^	1	1	1	1	1
Choline Chloride	0.5	0.5	0.5	0.5	0.5
Antimicrobial Agents	0.2	0.2	0.2	0.2	0.2
Antioxidants	0.1	0.1	0.1	0.1	0.1
Tea Polyphenols ^c^	0	0	0.01	0.02	0.04
Nutrient Composition ^d^ (%)					
Crude Protein	34.56	34.56	34.73	34.38	34.65
Crude Fat	5.44	5.44	5.49	5.55	5.61
Moisture	5.65	5.65	5.99	6.10	5.88

^a^. CON and CS groups fed basal diet without tea polyphenols; CSLTP, CSMTP, and CSHTP groups fed basal diet with 100, 200, and 400 mg/kg tea polyphenols. ^b^. The vitamin premix and mineral premix provided the following per kg of diets: VA 10,800 IU, VD3 4000 IU, VE 40 IU, VK3 3.4 mg, VB1 1.6 mg, VB2 12 mg, VB6 6 mg, VB12 0.05 mg, biotin 0.2 mg, folic acid 2 mg, niacin 50 mg, D-Calcium 25 mg, Fe 80 mg, Cu 100 mg, Mn 50 mg, Zn 90 mg, Co 1 mg, Se 0.17 mg, and I 0.15 mg. ^c^. Tea polyphenols were provided by the National Research Center of Engineering Technology for Utilization of Functional Ingredients from Botanicals, Hunan Agricultural University, Changsha, China. ^d^. Nutrient compositions were analyzed values.

**Table 2 animals-15-01983-t002:** Primes of antioxidant and lipid metabolism-related genes of hybrid crucian carp HCC2.

Gene	Functional Category	Forward Primer (5′-3′)	Reverse Primer (5′-3′)
*β-actin*	Reference gene	AAACGACCAACCCAAACC	GACGCTTCTGGAACGACTAA
*CAT*	Antioxidant enzyme	GCAAAGCCAAAGTGTTCG	CGCATCCCTGATAAAGAAG
*nrf2*	Antioxidant transcription factor	AATATGTCGCTATTAGTGTCG	AATCCAACGGAGGTGAAG
*SOD*	Antioxidant enzyme	CCGCACTACAACCCTCAT	CGGAGTATTCCCCAAACA
*GPX*	Antioxidant enzyme	CGACTCCGTGTCCTTGAT	GTTTATTTCGCCCTCTTC
*acox1*	Fatty acid β-oxidation enzyme	TGAGGACGCCTGGAACAA	CCGAGTGGACAGCCGTAT
*CPT1*	Fatty acid β-oxidation transporter	CGAGCAACAGATTCAGCG	AAGGGACTTGGCATAGCG
*LPL1*	Lipoprotein metabolism enzyme	CAGATCGCAGCATTGGG	CTCCGTAGCCGACCTTG
*dgat2*	Triglyceride synthesis enzyme	TCTCAGCCTTACACGACC	ACATCAGCAGCAAAGAGC

Abbreviation: *β-actin*: house-keeping gene, *CAT*: Catalase, *nrf2*: Nuclear Factor Erythroid 2-Related Factor 2, *SOD*: Superoxide Dismutase, *GPX*: Glutathione Peroxidase, *acox1*: Acyl-CoA Oxidase 1, *CPT1*: Carnitine Palmitoyl transferase 1, *LPL1*: Lipoprotein Lipase 1, *dgat2*, Diacylglycerol O-Acyltransferase 2.

**Table 3 animals-15-01983-t003:** Effects of dietary TPs on growth performance of hybrid crucian carp HCC2.

Items ^1^	CON	CS	CSLTP	CSMTP	CSHTP	CON vs. CS	CS vs. CS+TPs
						*p*-Value	*p*-Value
IBW (g)	2.03 ± 0.01	2.00 ± 0.00	2.00 ± 0.00	2.00 ± 0.00	2.00 ± 0.00	0.87	0.99
FBW (g)	14.37 ± 0.35 *	9.00 ± 0.84 ^b^	12.54 ± 0.93 ^a^	10.80 ± 0.26 ^ab^	9.29 ± 1.30 ^b^	<0.01	0.09
WGR (%)	609.32 ± 13.64 *	350.77 ± 42.28 ^b^	526.49 ± 46.09 ^a^	439.25 ± 11.98 ^ab^	364.07 ± 64.77 ^b^	<0.01	0.09
SGR (%/d)	3.26 ± 0.09 *	2.51 ± 0.30 ^b^	3.06 ± 0.27 ^a^	2.81 ± 0.08 ^ab^	2.56 ± 0.46 ^b^	<0.01	0.09
FCR	1.45 ± 0.01 *	1.80 ± 0.1	1.52 ± 0.05	1.62 ± 0.02	1.79 ± 0.13	0.03	0.14
SR (%)	100.00 ± 0.00	99.80 ± 0.2	99.67 ± 0.24	99.87 ± 0.07	99.80 ± 0.12	0.37	0.86
CF (g/cm^3^)	3.56 ± 0.05	3.51 ± 0.12 ^a^	3.42 ± 0.05 ^a^	3.17 ± 0.04 ^b^	3.36 ± 0.11 ^ab^	0.75	0.03

Data are shown as the AVG ± SEM (*n* = 3). The asterisk (*) as a superscript indicates a significant difference between CON and CS (*p* < 0.05) groups. Different superscript letters signify a significant difference between CS and CSLTP, CSMTP, and CSHTP groups (*p* < 0.05). ^1^ Abbreviation: IBW: initial body weight, FBW: final body weight, WGR: weight gain rate, SGR: specific growth rate, FCR: feed conversion efficiency, SR: survival rate, CF: condition factor.

**Table 4 animals-15-01983-t004:** Effects of dietary TPs on serum biochemical indices of hybrid crucian carp HCC2.

Items ^1^	CON	CS	CSLTP	CSMTP	CSHTP	CON vs. CS	CS vs. CS+TPs
						*p*-Value	*p*-Value
LDH (U/L)	538.00 ± 6.08 **	721.00 ± 14.93 ^a^	625.67 ± 19.64 ^b^	456.33 ± 4.98 ^c^	330.67 ± 11.57 ^d^	<0.01	<0.01
GLU (mmol/L)	16.30 ± 0.26 *	10.47 ± 0.60 ^a^	7.90 ± 0.29 ^b^	9.37 ± 0.23 ^a^	6.30 ± 0.35 ^c^	<0.01	<0.01
LACT (mmol/L)	9.96 ± 0.25	9.27 ± 0.27 ^a^	8.24 ± 0.32 ^b^	8.09 ± 0.16 ^b^	7.62 ± 0.20 ^b^	0.06	0.01
TG (mmol/L)	2.79 ± 0.08 **	4.37 ± 0.06 ^b^	3.81 ± 0.07 ^c^	5.39 ± 0.11 ^a^	4.28 ± 0.19 ^b^	<0.01	<0.01
TC (mmol/L)	7.32 ± 0.22	6.64 ± 0.19	7.46 ± 0.40	6.61 ± 0.31	7.33 ± 0.26	0.08	0.16
NEFA (mmol/L)	0.93 ± 0.02	1.31 ± 0.19 ^a^	1.60 ± 0.17 ^a^	1.55 ± 0.15 ^a^	0.67 ± 0.05 ^b^	0.12	0.01
LDL-C (mmol/L)	1.03 ± 0.10 *	0.55 ± 0.05	0.67 ± 0.06	0.53 ± 0.03	0.56 ± 0.01	0.02	0.16
HDL-C (mmol/L)	0.58 ± 0.01	0.5 ± 0.03 ^ab^	0.57 ± 0.03 ^a^	0.4 ± 0.02 ^b^	0.46 ± 0.02 ^ab^	0.14	0.01
ALB (g/L)	9.17 ± 0.20	9.93 ± 0.15	13.30 ± 1.97	10.40 ± 0.81	11.50 ± 0.31	0.06	0.2
ALT (U/L)	22.50 ± 2.10 *	14.93 ± 0.79 ^b^	10.87 ± 1.39 ^a^	16.83 ± 0.46 ^b^	21.80 ± 0.55 ^a^	0.03	<0.01
AST (U/L)	460.33 ± 3.38 *	378.67 ± 17.68 ^ab^	425.67 ± 29.04 ^a^	339.33 ± 3.18 ^b^	349.67 ± 12.60 ^b^	0.01	0.04
ALP (U/L)	18.90 ± 0.59 *	13.67 ± 0.33 ^b^	23.33 ± 2.73 ^a^	14.57 ± 0.30 ^b^	20.00 ± 0.58 ^a^	<0.01	<0.01

Data are shown as the AVG ± SEM (*n* = 3). The asterisk (*) as a superscript indicates a significant difference between CON and CS (*p* < 0.05) groups, while the double asterisk (**) as a superscript denotes a highly significant difference between CON and CS (*p* < 0.01) groups. Different superscript letters signify a significant difference between CS and CSLTP, CSMTP, and CSHTP groups (*p* < 0.05). ^1^ Abbreviation: LDH: Lactate Dehydrogenase, GLU: Glucose, LACT: Lactate, TG: Triglycerides, TC: Total Cholesterol, NEFA: Non-Esterified Fatty Acids, LDL-C: Low-Density Lipoprotein Cholesterol, HDL-C: High-Density Lipoprotein Cholesterol, ALB: Albumin, ALT: Alanine Aminotransferase, AST: Aspartate Aminotransferase, ALP: Alkaline Phosphatase.

**Table 5 animals-15-01983-t005:** Effects of dietary TPs on antioxidant capacity of hybrid crucian carp HCC2.

Items ^1^	CON	CS	CSLTP	CSMTP	CSHTP	CON vs. CS	CS vs. CS+TPs
						*p*-Value	*p*-Value
**Serum**							
T-AOC, μmol/mL	0.57 ± 0.06 *	0.71 ± 0.09	0.74 ± 0.08	0.65 ± 0.07	0.66 ± 0.06	0.03	0.48
SOD, U/mL	5.28 ± 0.44	5.73 ± 0.63 ^c^	8.46 ± 0.52 ^a^	7.69 ± 0.81 ^b^	7.89 ± 0.72 ^ab^	0.58	0.02
CAT, U/mL	26.3 ± 2.23	28.2 ± 2.32 ^b^	42.0 ± 3.45 ^a^	39.5 ± 3.61 ^a^	38.0 ± 4.01 ^a^	0.44	0.04
GSH, μg/mL	9.92 ± 0.97 **	5.49 ± 0.58 ^c^	8.33 ± 0.74 ^b^	11.85 ± 1.31 ^a^	9.28 ± 0.95 ^ab^	<0.01	<0.01
**Liver**							
T-AOC, μmol/g prot	4.65 ± 0.39	4.52 ± 0.42 ^a^	4.69 ± 0.36 ^a^	4.63 ± 0.38 ^a^	3.83 ± 0.45 ^b^	0.33	0.11
SOD, U/mg prot	111.8 ± 9.8	131.4 ± 10.3 ^ab^	105.9 ± 10.2 ^b^	141.5 ± 11.7 ^a^	128.3 ± 12.6 ^ab^	0.06	0.09
CAT, U/mg prot	252.02 ± 20.58	259.65 ± 21.53 ^b^	212.84 ± 22.14 ^c^	276.52 ± 22.88 ^ab^	305.10 ± 24.82 ^a^	0.24	0.08
GSH, μg/mg prot	1021.86 ± 66.20 *	951.12 ± 48.23 ^b^	988.73 ± 65.22 ^ab^	1060.66 ± 69.69 ^a^	1011.09 ± 77.47 ^ab^	0.04	0.16

Data are shown as the AVG ± SEM (*n* = 3). The asterisk (*) as a superscript indicates a significant difference between CON and CS (*p* < 0.05) groups, while the double asterisk (**) as a superscript denotes a highly significant difference between CON and CS (*p* < 0.01) groups. Different superscript letters signify a significant difference between CS and CSLTP, CSMTP, and CSHTP groups (*p* < 0.05). ^1^ Abbreviation: T-AOC: total antioxidant capacity; SOD: superoxide dismutase, CAT: catalase, GSH: glutathione.

## Data Availability

The data presented in this study are available on request from the corresponding author.

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
