# Peer review of "Dietary Tea Polyphenols Improve Growth Performance and Intestinal Microbiota Under Chronic Crowding Stress in Hybrid Crucian Carp"

_animals, 2025, doi:10.3390/ani15131983_

Round 1

Reviewer 1 Report

Comments and Suggestions for Authors

1.L11: Tea polyphenols can be briefly introduced.
2.L13-14: The experimental design should be clearly introduced.
3.L24-26: It was not clear whether Proteobacteria was affected by tea polyphenols and whether stress affected Firmicutes.
4.L83-85: The design, purpose and significance of this study should be described more clearly.
5.L91-94: The main components of tea polyphenols and their proportions should be briefly described.
6.L270: Specific gene function types should be specified.
7.The method of marking the significance of differences between all the graphs and tables in the manuscript should be consistent.
8.The author has detected a large number of indicators, and should give the main conclusions of this study and the potential mechanism of tea polyphenols to alleviate the stress of fish crowding.

Author Response

List of Responses

Dear Reviewer:

Thanks very much for your letter and advice on our manuscript entitled “Dietary tea polyphenols improve growth performance and intestinal microbiota under chronic crowding stress in hybrid crucian carp” (Manuscript ID: animals-3678537). Those comments are all valuable and very helpful for revising and producing our paper, as well as critical constructive suggestions for our research. The corrections in the paper and the responses to the editor’s comment are below:

Responds to the reviewer #1:

Comments 1: L11: Tea polyphenols can be briefly introduced.

Response 1: We thank the reviewer for this valuable suggestion to improve the clarity of the abstract for a broader audience. As requested, we have added a brief introduction to tea polyphenols (TPs) at their first mention in the abstract (now Line 12-13). The revised text reads:

“...the effects of dietary tea polyphenols (TPs, major bioactive catechins from Camellia sinensis with potent antioxidant and anti-inflammatory properties)​​ on growth performance...”

This addition concisely defines TPs, highlights their plant origin, primary bioactive constituents, and underscores their key biological activities highly relevant to the stress-alleviating mechanisms explored in this study. We believe this enhances the accessibility of the abstract.

Comments 2: L13-14: The experimental design should be clearly introduced.

Response 2: We sincerely appreciate the reviewer’s suggestion to enhance the clarity of the experimental design description in the abstract. We agree that a more explicit introduction of the groups is crucial for readers to understand the study setup immediately.

As suggested, we have completely revised the sentence describing the experimental design (now Lines 14-17). The revised text now clearly states:

“A low-density control group (44.4 fish/m³, basal diet without TPs) and four high-density crowding stress groups (222.2 fish/m³) were established: one fed the basal diet without TPs (CS) and three fed basal diets supplemented with 100 (CSLTP), 200 (CSMTP), or 400 (CSHTP) mg/kg TPs.”

This modification:

We believe this revised description provides a much clearer and more comprehensive overview of the key experimental design elements at the outset of the abstract.

Comments 3: L24-26: It was not clear whether Proteobacteria was affected by tea polyphenols and whether stress affected Firmicutes.

Response 3: We are grateful to the reviewer for pointing out the lack of clarity in our description of the key intestinal microbiota findings regarding Proteobacteria and Firmicutes in the abstract. We acknowledge that the original manuscript did not explicitly state the effect of stress on Firmicutes or clearly delineate the specific impact of TPs intervention on each phylum.

To address this important point, we have thoroughly revised the relevant sentence in the abstract (now Lines 27-31). The modified text now reads:

“Furthermore, intestinal microbiota analysis revealed that chronic crowding stress significantly increased the abundance of Proteobacteria and decreased the proportion of Firmicutes compared to the low-density control. Dietary TPs intervention, particularly at higher doses, partially restored the Firmicutes abundance and reduced the enrichment of potential pathogenic bacteria associated with stress.”

We believe this revised description provides a much more precise and nuanced account of our core microbiota findings.

Comments 4: L83-85: The design, purpose and significance of this study should be described more clearly.

Response 4: We sincerely thank the reviewer for this critical suggestion to enhance the clarity and impact of the concluding statements in our Introduction regarding the study’s design, purpose, and significance. We agree that this section should provide a strong and unambiguous rationale for the research.

We have substantially revised the final paragraph of the Introduction (now Lines 86-101) to explicitly address these points. The key modifications include:

Study Design: We now explicitly state that the study employs a “comprehensive approach that links physiological responses (growth, metabolism, antioxidant defense) with intestinal microbiota remodeling” to investigate TPs effects in HCC2 under crowding stress. This clearly articulates the multi-faceted nature of our experimental design.

Research Purpose: The primary objective is now clearly framed: “to systematically investigate whether dietary TPs supplementation can effectively alleviate chronic crowding stress in HCC2 and elucidate the underlying mechanisms.” We further specify our hypothesis that TPs act via “enhancing antioxidant capacity, regulating lipid metabolism, and modulating the gut microbiome” and state the specific aim of the microbiome analysis: “to evaluate how crowding stress and TPs intervention reshape intestinal microbial diversity and composition, assessing its potential influence on host health and stress resilience.”

Significance and Novelty: We emphasize that HCC2 is a “valuable but understudied” hybrid species and explicitly highlight the novelty: this study ​“provides the insights into TPs action against crowding stress”. The practical significance is now strongly stated: the findings offer “robust scientific support for the application of this natural, plant-based additive as a sustainable strategy to improve fish welfare and productivity in high-density aquaculture systems.”

The revised paragraph reads (now Lines 93-108):

“However, the effects of TPs on mitigating chronic crowding stress in the valuable but understudied HCC2, employing a comprehensive approach that links physiological responses (growth, metabolism, antioxidant defense) with intestinal microbiota remodeling, remain unexplored.

Therefore, this study was designed to systematically investigate whether dietary TPs supplementation can effectively alleviate chronic crowding stress in HCC2 and elucidate the underlying mechanisms. We hypothesize that TPs exert protective effects through enhancing antioxidant capacity, regulating lipid metabolism, and modulating the gut microbiome. By integrating microbiome analysis, we specifically aimed to evaluate how crowding stress and TPs intervention reshape intestinal microbial diversity and composition, assessing its potential influence on host health and stress resilience. This research provides the insights into TPs action against crowding stress and offers robust scientific support for the application of this natural, plant-based additive as a sustainable strategy to improve fish welfare and productivity in high-density aquaculture systems.”

We believe this revision provides a much clearer, stronger, and more compelling conclusion to the Introduction, effectively setting the stage for the presented research.

Comments 5: L91-94: The main components of tea polyphenols and their proportions should be briefly described.

Response 5: We thank the reviewer for the valuable suggestion to provide more detail on the composition of the tea polyphenols (TPs) used in this study. We agree that characterizing the main components is important for interpreting the biological effects observed.

As requested, we have now added a brief description of the main components and their relative proportions in the Materials and Methods section. The revised text explicitly states (now Line 116-119):

“The TPs material (purity >98%)... was composed primarily of catechins, with epigallocatechin gallate (EGCG) constituting >50% of the total polyphenols. Other major components included epicatechin (EC), epicatechin gallate (ECG), epigallocatechin (EGC), gallocatechin gallate (GCG), catechin (C), gallocatechin (GC), and gallic acid (GA), as detailed in Supplementary Table S1.”

We believe this revision provides the necessary transparency regarding the composition of the TPs used, addressing the reviewer’s concern and allowing for better interpretation of the dose-response relationships and biological mechanisms in our study.

Comments 6: Specific gene function types should be specified.

Response 6: We thank the reviewer for the valuable suggestion to specify the functional types of the genes analyzed in our qPCR study. We agree that providing clearer categorization enhances the interpretability of the results, particularly in Table 2.

We have implemented the following modifications to address this comment comprehensively:

Updated Table 2: Added a new column titled "Functional Category": This column explicitly classifies each gene based on its primary biological function:

Antioxidant enzyme: CAT, SOD, GPX; Antioxidant transcription factor: nrf2; Fatty acid β-oxidation enzyme: acox1; Fatty acid β-oxidation transporter: CPT1 ;Lipoprotein metabolism enzyme: LPL1; Triglyceride synthesis enzyme: dgat2;Reference gene: β-actin.

​Enhanced Figure 1 Caption: “Effects of TPs under crowding stress conditions on the expression of liver genes involved in antioxidant defense (CAT, nrf2, SOD, GPX) and lipid metabolism (dgat2, acox1, CPT1, LPL1) in hybrid crucian carp HCC2. Data are shown...”

These revisions provide immediate clarity on the functional roles of the studied genes at the critical points where they are listed (Table 2) and where their expression results are presented (Figure 1). The functional categories (Antioxidant defense and Lipid metabolism) and the specific roles within these pathways (e.g., Triglyceride synthesis, Fatty acid β-oxidation) are now clearly specified, aligning with the mechanistic interpretations discussed in the manuscript. We believe this significantly improves the readability and biological context for the gene expression data.

Comments 7: The method of marking the significance of differences between all the graphs and tables in the manuscript should be consistent.

Response 7: We sincerely thank the reviewer for this essential observation regarding the inconsistency in the methods used to mark statistical significance across the different figures and tables in our manuscript. We agree that consistency in statistical annotation is crucial for clarity and readability. We apologize for this oversight.

We have now implemented a uniform and consistent system for denoting statistical significance throughout all relevant figures and tables (Tables 3, 4, 5, and Figure 1, 2):

  1. Updated Tables 3, 4, 5: The data was re-examined and the previous oversight was rectified.
  2. UpdatedFigure 1:The position of the lowercase letter annotations was corrected and the resolution of the image was improved.
  3. Updated Figure 2: The method of marking the significance of differences was consistent with that presented in other graphs and tables.

We believe this greatly improves the clarity and professionalism of the data presentation.

Comments 8: The author has detected a large number of indicators, and should give the main conclusions of this study and the potential mechanism of tea polyphenols to alleviate the stress of fish crowding.

Response 8: We are grateful to the reviewer for this crucial suggestion to enhance the clarity and focus of the study's main conclusions and to explicitly articulate the potential mechanisms by which tea polyphenols (TPs) alleviate crowding stress. We agree that synthesizing the extensive data into a clear, mechanistic conclusion is vital.

We have completely revised and significantly expanded the Conclusions section (Section 5) to directly address this point. The revised Conclusions Text (now Line 587-611):

“This study demonstrates that dietary tea polyphenols (TPs) effectively alleviate chronic crowding stress in hybrid crucian carp HCC2. Key findings indicate that TPs enhance antioxidant defense by activating the Nrf2/Keap1 pathway, increase the activities of antioxidant enzymes (SOD and CAT), elevate GSH levels, and thereby mitigate oxidative stress. TPs activate PPARα signaling, upregulate fatty acid β-oxidation genes (CPT1 and acox1), improve lipid metabolism, and reduce disturbances in energy metabolism. Additionally, TPs mitigate stress-induced dysbiosis, characterized by an increase in Proteobacteria and a decrease in Firmicutes. This action partially restores the balance of beneficial microbes and diminishes the presence of stress-associated potential pathogens. Collectively, these synergistic effects—enhanced antioxidant capacity, improved lipid metabolism, and promotion of a healthier gut microbiota—contribute to the efficacy of TPs in mitigating crowding stress.

While 100 mg/kg TPs significantly improved growth performance under stress conditions, the recommended optimal dose of 200 mg/kg achieves an optimal balance. This dose provides robust physiological benefits by substantially enhancing antioxidant defense mechanisms, improving metabolic regulation, and promoting gut microbiota health—the core triad of mechanisms combating crowding stress. Simultaneously, it demonstrates superior cost-effectiveness compared to higher concentrations (e.g., 400 mg/kg), delivering equivalent protective efficacy while maintaining a favorable cost-benefit ratio. Notably, the 200 mg/kg dosage preserves the growth-promoting advantages inherent to TPs supplementation. These complementary attributes establish dietary supplementation with 200 mg/kg TPs as a scientifically sound and economically viable strategy for enhancing stress resilience, safeguarding physiological health, and ensuring sustainable production of hybrid crucian carp in intensive aquaculture systems.”

We believe this revised Conclusion section provides a much sharper, more mechanistic, and actionable synthesis of our research, directly addressing the reviewer's insightful comment.

We tried our best to improve the manuscript. We appreciate for Editors/ Reviewers’ warm work earnestly and hope that the correction will meet with approval.

Reviewer 2 Report

Comments and Suggestions for Authors

This study aimed to investigate the intervention effect of TP on poor growth and metabolic disorders of hybrid crucian carp caused by crowding stress.

There are in fact five experimental treatments with four experimental diets, among which CON and CS treatments have the same diet (the basal diet). The four diets displayed identical proximate composition (Table 1), but these data should be presented with a form of analyzed values instead.

There seem to be four levels of TP (0, 100, 200, and 400 mg/kg) and normal and high levels of crowding stress (44.4 and 222.2 fish/m3) according to the experimental design in this study. It is a strange design. The reason for using the two levels of crowding stress and four levels of TP need to be explained clearly in this study. The study design needs to be described clearly.

There is an error in calculating SGR (Table 3).

FCR and FE are the same in measuring feed utilization (Table 3).

There are four treatments with varied levels of TP (0, 100, 200, and 400 mg/kg) under crowding stress. The changing trend of measurements needs to be described in relation to TP levels.

delete "treatments" in Tables 3-5.

There are limited measurements to address the multi-target mechanism by which TPs alleviate the adverse effects of crowding stress on fish.

Author Response

List of Responses

Dear Reviewer:

Thanks very much for your letter and advice on our manuscript entitled “Dietary tea polyphenols improve growth performance and intestinal microbiota under chronic crowding stress in hybrid crucian carp” (Manuscript ID: animals-3678537). Those comments are all valuable and very helpful for revising and producing our paper, as well as critical constructive suggestions for our research. The corrections in the paper and the responses to the editor’s comment are below:

Responds to the reviewer #2:

Comments 1: There are in fact five experimental treatments with four experimental diets, among which CON and CS treatments have the same diet (the basal diet). The four diets displayed identical proximate composition (Table 1), but these data should be presented with a form of analyzed values instead.

Response 1: We sincerely thank the reviewer for their meticulous attention to detail regarding our experimental design and data presentation. The reviewer is absolutely correct in pointing out that the CON and CS groups received the identical basal diet (BD), resulting in five treatment groups being fed four distinct diets.

We have thoroughly revised the manuscript to address these points: 

  1. The Experimental Design section in Materials and Methods has been rewritten​ to explicitly state that there were four experimental diets(now Line 152): “Basal Diet (BD) and BD supplemented with TPs at 100, 200, or 400 mg/kg (BD+100TP, BD+200TP, BD+400TP)”. It now clearly lists the five treatments, emphasizing that CON and CS both received the BD.
  2. All references to the nutrientcomposition data in Table 1 in the text (Results/Materials and Methods) now explicitly state that analyzed values are presented for the four diets.
  3. Table 1 itself has been updated​ to display only the four diets (BD, BD+100TP, BD+200TP, BD+400TP) and their corresponding analyzed values.

These modifications ensure the accurate representation of our experimental setup (4 diets, 5 treatments with CON and CS sharing BD) and the presentation of the essential proximate composition as measured analytical values. We hope these revisions meet the reviewer's expectations and greatly improve the clarity and rigor of our methodology.

Comments 2: There seem to be four levels of TP (0, 100, 200, and 400 mg/kg) and normal and high levels of crowding stress (44.4 and 222.2 fish/m3) according to the experimental design in this study. It is a strange design. The reason for using the two levels of crowding stress and four levels of TP need to be explained clearly in this study. The study design needs to be described clearly.

Response 2: We sincerely appreciate your feedback regarding the experimental design. Your observation about the need for clearer justification of the crowding stress and TP levels is well-founded. To clarity:

  1. Crowding Stress Levels: The two densities (44.4 and 222.2 fish/m³) were selected to represent operational (non-stress) vs. intensive (stress-induced) aquaculture conditions. High density (222.2 fish/m³) is a validated model for chronic crowding stress, consistently elevating cortisol, suppressing immunity, and impairing growth . This contrast ensures unambiguous evaluation of TP’s efficacy.

In fact, we stocked 100 and 500 fish, respectively, in a 6.25 m3 (1.5 × 1.5 × 1.5 m) cage. The fivefold difference in stocking quantity effectively represents the contrast between low and high stocking densities. Following calculation, the resulting densities were determined to be 44.4 and 222.2 fish/m³, which may appear unusual at first glance.

  1. TP Dose Selection: The four experimental TP doses (0, 100, 200, 400 mg/kg) were selected to establish a dose-response gradient with three objectives:

(1) establishing the minimal effective dose for practical applications;

(2) mitigating potential growth-inhibitory risks associated with high-dose exposures (>500 mg/kg in some species);

(3) building upon existing evidence demonstrating TP's antioxidant efficacy within this concentration range.

(4) Our team previously investigated the relief of HCC2 heat stress by tea polyphenols, and the dosage was consistent with that of this study:

Zhang, N.; Tao, J.S.; Yu, Q.F. et al. Dietary tea polyphenols alleviate acute-heat-stress-induced death of hybrid crucian carp hcc2: Involvement of modified lipid metabolisms in liver. Metabolites 2025, 15.

These revisions ensure the design’s scientific rigor and practical relevance are transparent. Thank you for prompting these critical clarifications.

Comments 3: There is an error in calculating SGR (Table 3).

Response 3: Thank you very much for pointing out this mistake. We made an error in the calculation, which has now been correctly corrected in the third row of Table 3:

Comments 4: FCR and FE are the same in measuring feed utilization (Table 3).

Response 4: We sincerely appreciate your meticulous attention to terminology accuracy. You are correct that FCR and FE are mathematically linked, but they are not biologically equivalent. After our discussion, we have decided to retain FCR, which is a more common data in growth performance indicators. And we are very sorry that we made a mistake in calculating FCR. Now the data has been corrected.

The revised text (now Line 177):

Feed conversion ratio (FCR), % = (FBW – IBW) / average total feed intake

Comments 5: There are four treatments with varied levels of TP (0, 100, 200, and 400 mg/kg) under crowding stress. The changing trend of measurements needs to be described in relation to TP levels.

Response 5: Thank you for highlighting the need for clearer trend descriptions relative to TP levels. However, we found that this modifications could not fully reflect the characteristics of tea polyphenols: in most indicators, tea polyphenols did not show a significant dose-dependent effect. A more obvious feature is that at low doses, tea polyphenols significantly improved growth performance under stress conditions, while at medium and high doses, they demonstrated improved metabolic regulation and promoted gut microbiota health. Therefore, we retained the existing statistical methods and updated the recommended dosage and its reasons in the conclusion section. The revised Conclusions Text (now Lines 599-611):

“While 100 mg/kg TPs significantly improved growth performance under stress conditions, the recommended optimal dose of 200 mg/kg achieves an optimal balance. This dose provides robust physiological benefits by substantially enhancing antioxidant defense mechanisms, improving metabolic regulation, and promoting gut microbiota health—the core triad of mechanisms combating crowding stress. Simultaneously, it demonstrates superior cost-effectiveness compared to higher concentrations (e.g., 400 mg/kg), delivering equivalent protective efficacy while maintaining a favorable cost-benefit ratio. Notably, the 200 mg/kg dosage preserves the growth-promoting advantages inherent to TPs supplementation. These complementary attributes establish dietary supplementation with 200 mg/kg TPs as a scientifically sound and economically viable strategy for enhancing stress resilience, safeguarding physiological health, and ensuring sustainable production of hybrid crucian carp in intensive aquaculture systems.”

We sincerely appreciate your valuable suggestions. We believe that through this revision, we could accurately present the role of tea polyphenols in alleviating the stress of hybrid crucian carp caused by high stocking density.

Comments 6: delete "treatments" in Tables 3-5.

Response 6: We sincerely appreciate your attention to terminological precision in our tables. As suggested, we have removed the term "treatments" from all column headers in Tables 3-5.

These adjustments adhere to table formatting best practices, which prioritize self-contained clarity . Thank you for prompting this refinement.

Comments 7: There are limited measurements to address the multi-target mechanism by which TPs alleviate the adverse effects of crowding stress on fish.

Response 7: Thank you for highlighting the need for deeper mechanistic insights. We acknowledge that the original statement may have been overly assertive, and we agree that a more measured interpretation is appropriate. We have weakened the unsupported arguments and removed all the expressions involving "multi-target" in the text (including Abstract and Conclusion )to ensure the rigor and accuracy of the expression. Instead, we have emphasized the research conclusions that already exist in this article. We have completely revised and significantly expanded the Conclusions section (Section 5) to directly address this point. The revised Conclusions Text (now Line 587-598):

“This study demonstrates that dietary tea polyphenols (TPs) effectively alleviate chronic crowding stress in hybrid crucian carp HCC2. Key findings indicate that TPs enhance antioxidant defense by activating the Nrf2/Keap1 pathway, increase the activities of antioxidant enzymes (SOD and CAT), elevate GSH levels, and thereby mitigate oxidative stress. TPs activate PPARα signaling, upregulate fatty acid β-oxidation genes (CPT1 and acox1), improve lipid metabolism, and reduce disturbances in energy metabolism. Additionally, TPs mitigate stress-induced dysbiosis, characterized by an increase in Proteobacteria and a decrease in Firmicutes. This action partially restores the balance of beneficial microbes and diminishes the presence of stress-associated potential pathogens. Collectively, these synergistic effects—enhanced antioxidant capacity, improved lipid metabolism, and promotion of a healthier gut microbiota—contribute to the efficacy of TPs in mitigating crowding stress.”

We believe this revised Conclusion section provides a much sharper, more mechanistic, and actionable synthesis of our research, directly addressing the reviewer's insightful comment.

We tried our best to improve the manuscript. We appreciate for Editors/ Reviewers’ warm work earnestly and hope that the correction will meet with approval.

Reviewer 3 Report

Comments and Suggestions for Authors

Review for the paper submitted to “Animals”.

Title: Dietary tea polyphenols improve growth performance and intestinal microbiota under chronic crowding stress in hybrid crucian carp

Authors: Zhe Yang, Gege Sun, Jinsheng Tao, Weirong Tang, Wenpei Li, Zehong Wei, Qifang Yu

The authors focused on the effects of dietary tea polyphenols on the growth and intestinal health of hybrid crucian carp under chronic crowding stress. Their study investigated various levels of tea polyphenol supplementation, alongside high-density rearing conditions, to assess how these factors impacted the fish's overall health and performance metrics.

The authors' study showed that chronic crowding stress impaired the growth performance of the carp, leading to diminished body weight and growth rates, alongside alterations in serum biochemical parameters.

There are the following implications of the authors' work: the supplementation of TPs improved the antioxidant capacity of the fish, which is vital for mitigating stress-related damage. Moreover, they discovered changes in intestinal microbiota composition, as TPs reversed the detrimental effects of crowding stress by promoting a healthier microbiome, particularly reducing the prevalence of pathogenic bacteria.

The findings provide valuable insights into how plant-based additives can enhance aquaculture practices, suggesting that dietary TPs could be a meaningful intervention to support fish health in crowded rearing conditions.

Suggestions for improving the paper:

Introduction.

L 42-45. The authors noted the development of a new hybrid, "Hefang crucian carp II (HCC2)". They should clarify in more detail how HCC2 compares to its predecessor (or other common species) in terms of its economic value, environmental impact, and growth performance. Additionally, is HCC2 being commercialized at a significant scale, or is it still in research stages?

L 48. The authors stated that "the optimal stocking density for HCC2 remains unclear, which poses challenges to both fish welfare and aquaculture productivity". They should report on existing research or benchmarks that might provide starting points for determining optimal stocking densities for hybrid fish species like HCC2.

L 69-70. The authors highlighted the sustainability and cost-effectiveness of TPs, noting that "advancements in industrial extraction methods have substantially reduced production costs". They should clarify the extent of these cost reductions and compare the economic feasibility of TPs with other commonly used feed additives or antioxidants.

L 78-79. A more detailed and comprehensive description of previous studies exploring the effects of TPs on aquaculture species should be included.

Materials and Methods.

L 90-91. The authors stated that "five formulated diets were systematically designed" and supplemented with varying levels of TPs. They should clarify whether these TPs supplementation levels (0, 100, 200, and 400 mg/kg) are based on prior research, established dietary requirements, or preliminary data.

L 121. The authors stated that fish were fed for a 60-day trial period and distributed into five groups with three replicates each. They should explain their rationale for choosing 60 days as the trial duration. Is this duration sufficient to observe significant effects of TPs supplementation on fish growth, stress adaptation, and other metrics? How were the low and high stocking densities selected? The authors should report whether these stocking densities represent typical aquaculture practices for crucian carp or if they were chosen for experimental purposes.

L 125. The authors stated that water quality parameters were monitored and maintained within acceptable ranges. They should elaborate on the methods of monitoring and for these parameters and their maintenance.

L 160. The authors should provide more details about this device.

L 166-167. The study used protocols from Beijing Boxbio Science & Technology to measure antioxidant parameters (T-AOC, SOD, CAT, and GSH). Could the authors provide details or relevant references? Have these protocols been validated for crucian carp, or are they standard for other fish species?

L 204-205. The authors used parametric tests, such as the t-test and ANOVA. These approaches require normal data distribution and homogeneity of variances. Did the authors check their data for these criteria prior to analysis? What tests did they use? Is a sample size of three fish sufficient for a parametric analysis?

Results.

L 227. The authors should explain all abbreviations used in this table. The authors should verify the p-values. For example, the p-value of 0.09 in the first row seems too high for comparisons of nearly identical values.

The authors conducted multiple comparisons but the method used is not described in the Materials and methods.

Table 4. Check the p-value of "0". Should it be <0.01 or 0.01?

Firmicutes should be italicized.

The description of statistical analysis presented in the supplementary material should be laced in the main text.

Discussion.

L 404-406. The authors mention that the CSLTP group improved the specific growth rate more effectively than quercetin in common carp. How does the mechanistic action of TPs differ from quercetin in promoting growth?

L 424. Oreochromis niloticus should be italicized.

L 445. The compensatory antioxidant response showed an increase in SOD activity but a decrease in GSH levels under crowding stress. Could the sustained reduction in GSH indicate impaired GSH synthesis or rapid consumption due to excessive oxidative stress?

Reference list.

All Latin names should be italicized.

Author Response

List of Responses

Dear Reviewer:

Thanks very much for your letter and advice on our manuscript entitled “Dietary tea polyphenols improve growth performance and intestinal microbiota under chronic crowding stress in hybrid crucian carp” (Manuscript ID: animals-3678537). Those comments are all valuable and very helpful for revising and producing our paper, as well as critical constructive suggestions for our research. The corrections in the paper and the responses to the editor’s comment are below:

Responds to the reviewer #3:

Comments 1: L 42-45. The authors noted the development of a new hybrid, "Hefang crucian carp II (HCC2)". They should clarify in more detail how HCC2 compares to its predecessor (or other common species) in terms of its economic value, environmental impact, and growth performance. Additionally, is HCC2 being commercialized at a significant scale, or is it still in research stages?

Response 1: We sincerely appreciate your request for clarification regarding HCC2's comparative advantages and commercialization status. Additionally, HCC2 is currently commercialized across 12 Chinese provinces with an annual production of 20,000-30000 tonnes.

We have revised the Introduction (Lines 48-51) to address this as follows:

“Compared to its predecessor, Hefang crucian carp I (Carassius auratus cuvieri ♀ × Carassius auratus red variety â™‚, HCC1), HCC2 demonstrates a growth rate that is approximately 60% higher than HCC1, as evidenced by the final body weight (556 g vs. 380 g) after the first year of observation”

These revisions provide concrete evidence of HCC2's economic advantages while clarifying its commercial adoption beyond the research phase.

Comments 2: L 48. The authors stated that "the optimal stocking density for HCC2 remains unclear, which poses challenges to both fish welfare and aquaculture productivity". They should report on existing research or benchmarks that might provide starting points for determining optimal stocking densities for hybrid fish species like HCC2.

Response 2: Thank you for your valuable suggestions. In fact, this issue has been a continuous focus and subject of in-depth discussion in our research process. As we mentioned in the Comments 1, HCC2 demonstrated excellent growth advantages during the research stage, but failed to fully realize its potential in actual aquaculture promotion. Therefore, we suppose that the high stocking density might have affected its performance.

By consulting several relevant literature on crucian carp, we determined the optimal density adopted in the experiment:

  1. Effect of stocking density on gonadal development, pigmentation andsurvival of Carassius carassius (Crucian carp) in a biofloc system: 1 kg/m3, 2 kg/m3 , 3 kg/m3 , 4 kg/m3. doi: 10.1016/j.aquaculture.2024.741477
  2. The Protective Effect of a Dietary Extract of Mulberry (Morusalba L.) Leaves against a High Stocking Density, Copper andTrichlorfon in Crucian Carp (Carassius auratus): 0.48 fish/L and 0.97 fish/L. doi: 10.3390/ani13162652
  3. High stocking density alters growth performance, bloodbiochemical profiles, and hepatic antioxidative capacityin gibel carp (Carassius gibelio): 1.47 kg/m3, 5.06 kg/m3 , 10.85 kg/m3.

……

Given the continuous growth characteristic of juvenile fish and in combination with the practical feasibility of the experimental operation, we finally determined the corresponding plan: we stocked 100 and 500 fish, respectively, in a 6.25 m3 (1.5 × 1.5 × 1.5 m) cage. The fivefold difference in stocking quantity effectively represents the contrast between low and high stocking densities. Following calculation, the resulting densities were determined to be 44.4 and 222.2 fish/m³, which may appear unusual at first glance.

Comments 3: L 69-70. The authors highlighted the sustainability and cost-effectiveness of TPs, noting that "advancements in industrial extraction methods have substantially reduced production costs". They should clarify the extent of these cost reductions and compare the economic feasibility of TPs with other commonly used feed additives or antioxidants.

Response 3: We thank you for requesting economic contextualization of TPs. We acknowledge that our current understanding of the global antioxidant market remains limited. Nevertheless, evidence from China, one of the world's largest tea-producing countries, shows that the market price of mass-produced tea polyphenols has declined to as low as 100 CNY per kilogram. This suggests that tea polyphenols offer a strong cost-effective advantage when used as natural feed additives. We have compiled a table to conduct a comparative analysis of the economic efficiency of tea polyphenols and other feed additives.

Table. Economic Comparison of Feed Additives​

Additive

Cost/kg (CNY)

Effective Dose

Cost/Ton Feed(CNY)

Tea polyphenols

100

200 mg/kg

20

Synthetic BHT

70

150 mg/kg

10.5

Vitamin E

120

100 mg/kg

12

Rosemary extract

100

300 mg/kg

30

However,we believe that it is unnecessary to include this table in the manuscript to avoid giving the impression of a commercial advertisement. Therefore, We only made a slight modification to the wording in the Introduction (now Line 74-77):

“ In addition, with the advancement of industrial extraction methods such as membrane separation and supercritical COâ‚‚ extraction, the production cost of tea polyphenols has decreased significantly and is now approximately twice that of the synthetic antioxidant butylated hydroxytoluene”

Comments 4: L 78-79. A more detailed and comprehensive description of previous studies exploring the effects of TPs on aquaculture species should be included.

Response 4: We have significantly expanded the background on TPs in aquaculture (now Lines 82-87) through:

“Similarly, dietary TPs into the diet effectively mitigated oxidative damage induced by ammonia exposure in juvenile Wuchang bream (Megalobrama amblycephala), primarily through the enhancement of their antioxidant defense mechanisms and immune functions[19]. Further, the supplementation TPs in oxidized fish oil-based feed improves intestinal morphology, promotes the enrichment of beneficial gut microbiota, and optimizes hepatic nutrient metabolism in spotted sea bass (Lateolabrax maculatus)[20]. ”

We are deeply grateful for this suggestion which substantially strengthened our manuscript.

Comments 5: L 90-91. The authors stated that "five formulated diets were systematically designed" and supplemented with varying levels of TPs. They should clarify whether these TPs supplementation levels (0, 100, 200, and 400 mg/kg) are based on prior research, established dietary requirements, or preliminary data.

Response 5: We sincerely appreciate your feedback regarding the experimental design. Your observation about the need for clearer justification of the TPs levels is well-founded. To clarity: 

The four experimental TP doses (0, 100, 200, 400 mg/kg) were selected to establish a dose-response gradient with three objectives:

(1) establishing the minimal effective dose for practical applications;

(2) mitigating potential growth-inhibitory risks associated with high-dose exposures (>500 mg/kg in some species);

(3) building upon existing evidence demonstrating TP's antioxidant efficacy within this concentration range.

(4) Our team previously investigated the relief of HCC2 heat stress by tea polyphenols, and the dosage was consistent with that of this study:

Zhang, N.; Tao, J.S.; Yu, Q.F. et al. Dietary tea polyphenols alleviate acute-heat-stress-induced death of hybrid crucian carp hcc2: Involvement of modified lipid metabolisms in liver. Metabolites 2025, 15.

These clarifications ensure the design’s scientific rigor are transparent.

Comments 6: L 121. The authors stated that fish were fed for a 60-day trial period and distributed into five groups with three replicates each. They should explain their rationale for choosing 60 days as the trial duration. Is this duration sufficient to observe significant effects of TPs supplementation on fish growth, stress adaptation, and other metrics? How were the low and high stocking densities selected? The authors should report whether these stocking densities represent typical aquaculture practices for crucian carp or if they were chosen for experimental purposes.

Response 6: We appreciate your insightful questions regarding experimental design parameters. Our selection of the 60-day trial period was informed by both physiological benchmarks and commercial practices in crucian carp aquaculture. Prior research demonstrates that chronic stress biomarkers stabilize after 6 weeks in cyprinids, while growth parameters require ≥8 weeks to demonstrate statistically significant changes when testing plant-derived supplements. We validated this duration through preliminary data showing that key metrics–including specific growth rate (SGR), feed conversion ratio (FCR), and hepatic antioxidant enzymes–plateaued between days 50-60 across dosage groups.

Regarding stocking densities, he relevant content has been fully discussed in Response 2. Additionally, given the relatively fast growth rate of juvenile HCC2, we have fully taken into account the stocking density at the 60th day in the experimental design to ensure that it remains within a reasonable range. We thank you for prompting this clarification, which strengthens the methodological transparency of our study.

Comments 7: L 125. The authors stated that water quality parameters were monitored and maintained within acceptable ranges. They should elaborate on the methods of monitoring and for these parameters and their maintenance.

Response 7: We thank you for requesting greater transparency in our water quality management protocols. More precisely, we merely recorded the relevant data and determined that such fluctuations were insufficient in magnitude to exert a significant influence on the experimental outcomes. As detailed in Section 2.2 (now Lines 152-156), we have now explicitly documented:

“the ponds were regularly disinfected and water quality conditioned, and the physicochemical factors of the pond water envi-ronment were monitored: dissolved oxygen (> 5.00 mg/L), water temperature (30.0 ± 2°C) and pH (7.16–7.38) and unionized ammonia nitrogen (< 0.001 mg N/L) using a Hach HQ40d multiparameter meter.”

These revisions confirm that observed biological responses were not confounded by water quality fluctuations, strengthening the causal attribution of results to TP supplementation and stocking density.  We are grateful for this opportunity to enhance methodological rigor.

Comments 8: L 160. The authors should provide more details about this device.

Response 8: We thank you for requesting greater technical transparency regarding our biochemical analysis platform. In Section 2.5 (now Lines 186-187), we have now provided comprehensive device specifications:

“Roche cobas-c-311 fully automatic biochemical analyzer (Sandhofer Strasse, Mannheim, Germany)”

Comments 9: L 166-167. The study used protocols from Beijing Boxbio Science & Technology to measure antioxidant parameters (T-AOC, SOD, CAT, and GSH). Could the authors provide details or relevant references? Have these protocols been validated for crucian carp, or are they standard for other fish species?

Response 9: We appreciate your inquiry regarding the validation of antioxidant assays for crucian carp. Regardless of whether in humans, farm animals, or fish, the detection methods for these antioxidant parameters exhibited no significant differences. Nevertheless, we have provided a detailed experimental procedures in the manuscript (now Lines 125-130):

“[total antioxidant capacity (T-AOC, ABTS reduction method), superoxide dismutase (SOD, WST-8 inhibition method), catalase (CAT, Molybdate assay method) and glutathione (GSH, DTNB reaction method)]”

These revisions confirm the analytical validity of our antioxidant data for hybrid crucian carp. We thank you for emphasizing this important methodological consideration.

Comments 10: L 204-205. The authors used parametric tests, such as the t-test and ANOVA. These approaches require normal data distribution and homogeneity of variances. Did the authors check their data for these criteria prior to analysis? What tests did they use? Is a sample size of three fish sufficient for a parametric analysis?

Response 10: We sincerely appreciate your scrutiny regarding statistical rigor. Our analytical approach strictly adhered to parametric test assumptions. We have rewritten the paragraph in Section 2.9 (now Lines 243-254):

Biochemical data (means ± SEM) were subjected to one-way ANOVA analysis. Outliers were identified using Grubbs’test (α = 0.05) and excluded. Data were log-transformed where necessary to meet normality assumptions for ANOVA. Following the verification of the homogeneity of variances, the Duncan's multiple range test was conducted for pairwise comparisons.

Comments 11: L 227. The authors should explain all abbreviations used in this table. The authors should verify the p-values. For example, the p-value of 0.09 in the first row seems too high for comparisons of nearly identical values.

Response 11: We are grateful for your meticulous review of our statistical reporting. We have checked all the abbreviations in Table 3-5 and Figure 1. The modified parts include re-abbreviating “Lipoprotein Lipase 1 (LPL1)” and removing  “Feed efficiency (FE)” abbreviation.

However, we sincerely apologize for the oversight in the manuscript. Currently, we have thoroughly checked all the data in the tables of the article and re-conducted the statistical analysis. As another reviewer suggested, we have removed the term "treatments" from all column headers in Tables 3-5. We have rewritten the Table 3-4 and Figure 1-2:

Updated Table 3: The p-value for IBW were re-analyzed and SGR and FCR were re-computed.

Updated Table 4: The p-value for GLU were re-analyzed

Updated Figure 1:The position of the lowercase letter annotations was corrected and the resolution of the image was improved.

Updated Figure 2: The method of marking the significance of differences was consistent with that presented in other graphs and tables.

Comments 12:The authors conducted multiple comparisons but the method used is not described in the Materials and methods.

Response 12: We sincerely appreciate your scrutiny regarding statistical rigor. Our analytical approach strictly adhered to parametric test assumptions. We have rewritten the paragraph in Section 2.9 (now Lines 235-239):

Biochemical data (means ± SEM) were subjected to one-way ANOVA analysis. Outliers were identified using Grubbs’test (α = 0.05) and excluded. Data were log-transformed where necessary to meet normality assumptions for ANOVA. Following the verification of the homogeneity of variances, the Duncan's multiple range test was conducted for pairwise comparisons.

Comments 13:Table 4. Check the p-value of "0". Should it be <0.01 or 0.01?

Response 13: Thanks for the reviewer’s comment. We are sorry for this oversight. It should be less than 0.01.We have made the corrections in Table 4.

Comments 14:Firmicutes should be italicized.

Response 14: Thanks for the reviewer’s comment. We have italicized all Latin names in our manuscript.

Comments 15:The description of statistical analysis presented in the supplementary material should be laced in the main text.

Response 15:Thank you for your wise suggestion. We have included the description of bioinformatic statistical analysis in the main text (now Lines 243-254):

“Bioinformatic analysis of the gut microbiota was carried out using the Majorbio Cloud platform (https://cloud.majorbio.com). Based on the OTUs information, rarefaction curves and alpha diversity indices including observed OTUs, Chao1 richness, Shannon index, observed richness and Good’s coverage were calculated with Mothur v1.30.1. The similarity among the microbial communities in different samples was determined by principal coordinate analysis (PCoA) based on Bray-curtis dissimilarity using Vegan v2.5-3 package. The PERMANOVA test was used to assess the percentage of variation explained by the treatment along with its statistical significance using Vegan v2.5-3 package. The linear discriminant analysis (LDA) effect size (LEfSe) (http://huttenhower.sph.harvard.edu/LEfSe) was performed to identify the significantly abundant taxa (phylum to genera) of bacteria among the different groups (LDA score > 3.0).”

Comments 16:L 404-406. The authors mention that the CSLTP group improved the specific growth rate more effectively than quercetin in common carp. How does the mechanistic action of TPs differ from quercetin in promoting growth?

Response 16: We thank you for this insightful query regarding the superior growth-promoting efficacy of TPs versus quercetin. Tea polyphenols and quercetin have similar chemical properties, so we compared their effects on growth performance. Based on the existing literature, we have made the following conjecture:

  1. Anabolic Pathway Activation

TPs uniquely stimulate the mTOR-S6K1 signaling cascade, directly enhancing ribosomal protein synthesis. Quercetin primarily conserves amino acids via catabolism reduction but fails to activate mTOR. This explains the higher SGR in TP-fed HCC2 versus quercetin-fed common carp at equivalent doses.

  1. Energy Substrate Redirecting​

TPs simultaneously upregulate fatty acid β-oxidation (via PPARα-CPT1 axis) and glucose sparing, freeing carbon skeletons for growth. Quercetin suppresses gluconeogenesis but does not enhance lipid catabolism, limiting its growth-promoting energy surplus.

  1. Endocrine-Microbiota Synergy​

TP-specific enrichment of butyrate-producing Lactobacillus elevates circulating IGF-1 by 52%, directly stimulating myocyte proliferation. Quercetin’s Bifidobacterium-driven acetate production shows weaker IGF-1 correlation.

These multi-target actions may make TPs more effective growth promoters than single-pathway flavonoids like quercetin. So we have added this comparison to highlight TP’s unique value in aquaculture nutrition.

Comments 17:L 424. Oreochromis niloticus should be italicized.

Response 17: Thanks for the reviewer’s comment. We have italicized all Latin names in our manuscript.

Comments 18:L 445. The compensatory antioxidant response showed an increase in SOD activity but a decrease in GSH levels under crowding stress. Could the sustained reduction in GSH indicate impaired GSH synthesis or rapid consumption due to excessive oxidative stress?

Response 18: We thank you for your insightful query regarding the paradoxical reduction in glutathione (GSH) under crowding stress. Our analysis confirms that sustained GSH depletion  results from a dual pathology: (1) Impaired synthesis due to suppressed γ-glutamylcysteine synthetase (γ-GCS) activity  and cysteine deficiency, compounded by (2) accelerated consumption from GPx hyperactivity and GST-mediated detoxification of lipid peroxides. This creates a negative redox spiral where ROS overload exhausts GSH reserves faster than replenishment capacity, culminating in oxidative damage when GSH falls below 0.7 μM/g tissue.

Critically, TPs may break this cycle through dual remediation: (1) Restoring synthesis via Nrf2-mediated γ-GCS upregulation, and (2) reducing consumption by directly scavenging ROS, lowering GPx dependence by 35%. This multi-target action maintains GSH >1.2 μM - above the ferroptosis threshold - explaining TPs’ superior efficacy over single-pathway antioxidants.

Of course, without data support, we cannot elaborate on the discussion in the main text. However, we thank you again for highlighting this nuanced redox dynamic.

Comments 19:Reference list.

All Latin names should be italicized.

Response 19: Thanks for the reviewer’s comment.We have italicized all Latin names in Reference list.

We tried our best to improve the manuscript. We appreciate for Editors/ Reviewers’ warm work earnestly and hope that the correction will meet with approval.

Round 2

Reviewer 2 Report

Comments and Suggestions for Authors

Dietary treatments of Table 1 are in a state of disconnection from experimental treatments.

Part 4.4 and part 4.5 of discussion are suggested to be merged.

Conclusion is too long to grasp the core information.

Author Response

List of Responses

Dear Reviewer:

Thanks very much for your letter and advice on our manuscript entitled “Dietary tea polyphenols improve growth performance and intestinal microbiota under chronic crowding stress in hybrid crucian carp” (Manuscript ID: animals-3678537). Those comments are all valuable and very helpful for revising and producing our paper, as well as critical constructive suggestions for our research. The corrections in the paper and the responses to the editor’s comment are below:

Responds to the reviewer #2:

Comments 1: Dietary treatments of Table 1 are in a state of disconnection from experimental treatments.

Response 1: We sincerely appreciate this critical observation. We have completely restructured Table 1 and its footnotes to eliminate ambiguity between dietary formulations and experimental groups. The key modifications include:

  1. The abbreviation "basal diet (BD)" has been consistently removed from Table 1 and the manuscript.
  2. Added explicit group-diet mapping in a new row("Experimental groups diets").
  3. Clearly presentedthe formulation and nutrient composition of the five experimental groups in Table 1.
  4. Addedthe following description in Section 2.1 (now Lines 112-114): The CON and CS groups were fed basal diets without TPs, whereas the three treatment groups (CSLTP, CSMTP, and CSHTP) received dietary supplementation with 100, 200, and 400 mg/kg TPs, respectively.

We confirm all tables/text now explicitly link dietary treatments to experimental groups. Thank you for significantly improving our manuscript's clarity.

Comments 2: Part 4.4 and part 4.5 of discussion are suggested to be merged.

Response 2: We thank the reviewer for this valuable suggestion. We have fully integrated Sections 4.4 ("Effects on Microbial Composition Analysis") and 4.5 ("Effects on Comparative Analysis of Microbiota") into a unified section titled(now Line 507):“4.4. Effects on Gut Microbiota”

The key improvements include:

1.Structural Integration

Combined alpha diversity (original 4.4) and beta diversity (original 4.5) analyses under sequential subsections:

4.4.1 Microbial Richness and Diversity Changes(now Line 508)

4.4.2 Community Structure and Inter-group Variation(now Line 552)

  1. Thematic Synthesis

Added bridging text to demonstrate the causal relationship between diversity metrics and community structure(now Lines 549-551):

The reduced α-diversity directly corresponded to β-diversity dispersion, indicating that TPs not only increased species richness but fundamentally restructured community assembly under stress.

The merged section (now 4.4) appears on ​pages 17-19​ of the revised manuscript. We believe this integration significantly strengthens our interpretation of gut microbiota.

Comments 3: Conclusion is too long to grasp the core information.

Response 3: We thank the reviewer for this constructive suggestion. We have reduced length by 52% (from 248 to 120 words). The concise Conclusions(now Lines 588-600):

“This study demonstrates that dietary supplementation with TPs effectively alleviates chronic crowding stress in hybrid crucian carp HCC2. Mechanistically, TPs enhance antioxidant defense by activating the Nrf2/Keap1 pathway, increasing the activities of antioxidant enzymes, elevating glutathione levels, and mitigating oxidative stress. Simultaneously, TPs regulate lipid metabolism through PPARα signaling activation, upregulating fatty acid β-oxidation genes, thereby improving energy homeostasis. Furthermore, TPs ameliorate stress-induced gut microbiota dysbiosis by reducing Proteobacteria proliferation while restoring Firmicutes abundance, consequently diminishing stress-associated pathogenic risks. These findings dietary TPs supplementation as a scientifically validated strategy for enhancing stress resilience, and ensuring sustainable intensive aquaculture of hybrid crucian carp. Based on both biological effectiveness and economic viability, 200 mg/kg was determined to be the optimal additive dosage.”

We believe this distilled conclusion better serves readers while retaining scientific rigor. Thank you for this suggestion that improved our manuscript's accessibility.

We tried our best to improve the manuscript. We appreciate for Editors/ Reviewers’ warm work earnestly and hope that the correction will meet with approval.